# Contrastive Subgroups: Discovering Where Two Populations Differ, and Why

## Abstract

Given data from two distinct populations, a contrastive subgroup describes a subset of individuals from both groups who, despite sharing similar characteristics, exhibit significant differences in a target outcome. For example, we want to identify subsets of patients who respond differently to a treatment compared to a control group, or uncover disparities between protected and unprotected groups in fairness analysis. In this work, we formalize the notion of contrastive subgroups and propose a general optimization objective to discover them. To make these discovered subgroups actionable, we provide conditions under which the discovered subgroups allow to make causal inferences. We introduce SubCon, a gradient-based method to discover contrastive subgroups and evaluate it on both synthetic and real-world datasets. The results confirm that our method effectively identifies subgroups that expose significant, informative differences in real-world datasets.

## 1 Introduction

Understanding the differences between two distinct populations is a fundamental challenge in data analysis, with significant implications across domains. In clinical trials, researchers aim to identify which patients benefit most from a treatment compared to a control group (Lipkovich et al., 2011). In fairness analysis, a key task is to determine if a protected group faces different outcomes than an unprotected group, even when their observable characteristics are similar (Dwork et al., 2012).

However, existing approaches have limitations: In machine learning fairness, the focus lies on training models that are fair across a protected group and an unprotected group (Kearns et al., 2018; Shui et al., 2022). These methods provide notions to quantify disparity between two populations, but they leave open where that disparity arises and why their distributions diverge. In clinical trials, subgroups are often defined a priori to preserve statistical power (Cook et al., 2004). Methods that do discover subgroups focus on a particular data setting, such as randomized control trials (Foster et al., 2011; Ballarini et al., 2018) and observational trials (Athey & Imbens, 2016).

In this paper, we propose a novel task that fills this gap and is applicable to various domains: discovering **contrastive subgroups**. Taking inspiration from subgroup discovery (Atzmueller, 2015), where the goal is to identify a sub-population with exceptional behavior, we extend this concept to focus on the differences between two populations. We aim to discover subgroups of individuals who share similar characteristics, but where the local distribution of a target significantly varies between the two populations. This allows us to pinpoint precisely where and why two populations diverge, providing a more nuanced understanding of the differences between them.

To this end, we propose SubCon, a method that discovers contrastive subgroups in tabular data. The primary contributions of this work are:

1. We formalize the notion of contrastive subgroups and define an objective that contains the three desiderata of exceptionality, generality, and covariate independence.

2. We investigate under which conditions contrastive subgroups accurately reflect the causal effect of membership in either population on the target variable.

3. We present a continuous-optimization based method to discover contrastive subgroups.

Our experimental results show that SUBCON effectively discovers interpretable subgroups that exhibit significant differences in the target variable.

## 2 RELATED WORK

Defining and quantifying disparity between a protected and unprotected group is the focus of fairness in machine learning. For example, the notion of demographic parity (Dwork et al., 2012) postulates that the marginal distribution of the target variable should be similar across protected groups. There exist similar definitions for subgroups, but the existing work focuses on training classifiers that are compliant (Kearns et al., 2018; Martinez et al., 2021; Shui et al., 2022), rather than discovering disparity in an existing dataset.

**Subgroup Discovery** on tabular databases is a well-established task (Atzmueller, 2015). The goal is to discover a subset of individuals that exhibit exceptional behavior, e.g. a subgroup that has a higher survival rate than the overall population. The exceptionality of a subgroup is here quantified as the deviation from the overall population, e.g. by comparing means (Lemmerich et al., 2016), correcting for dispersion (Boley et al., 2017), or comparing the entire distribution (Xu et al., 2024).

Kalofolias et al. (2017) extend subgroup discovery to a protected attribute. They aim to minimize the effect of the sensitive attribute on the target variable by ensuring that the distribution of a sensitive attribute, e.g. the ratio of males to females, is similar in the subgroup to the overall population. However, they still only search for subgroups that diverge from the overall population.

Contrast set mining (Bay & Pazzani, 2001, CSM) is another closely related topic. The goal of CSM is to find feature characteristics that occur significantly more often in a particular group. However, CSM differs from contrastive subgroups in a crucial way: CSM's aim is to find characteristics of a particular group, e.g. discovering feature characteristics that distinguish men from women. On the other hand, we are interested in differences regarding a third variable, e.g. finding exceptional differences in the survival rate between men and women.

**Treatment Effect Estimation** is a field where it is a key challenge to discover interesting subgroups. In general, the goal is to estimate the effect of a treatment compared to a control group (Imbens & Rubin, 2015). For randomized trials, where groups are randomly assigned, treatment effect subgroups are discovered using regression trees (Lipkovich et al., 2011), trees augmented with local linear models (Seibold et al., 2016), or using Lasso regression (Ballarini et al., 2018).

For observational trials, where the treatment is not randomly assigned, the task is more challenging. Most methods focus on the faithful estimation of heterogeneous treatment effects, for which they employ non-interpretable ML models (Künzel et al., 2019) and neural networks (Shalit et al., 2017). Athey & Imbens (2016) introduce honest causal trees, where every leaf node represents a subgroup with an unbiased estimated treatment effect. However, it and the subsequent causal forest method (Wager & Athey, 2018) place restrictive assumptions on the causal model, and focus on estimating the treatment effect, rather than discovering interesting subgroups.

In general, the notion of contrastive subgroups is an often occurring task in many domains. However, as of now, there does not exist a general definition or method to discover them in a principled way.

## 3 PRELIMINARIES

We consider a dataset of $n$ individuals $(x^{(i)}, a^{(i)}, y^{(i)})$, $i \in \{1, \ldots, n\}$, where $y$ is the **target variable**, $x$ represents the **descriptive features**, and $a \in \{0, 1\}$ is the **binary attribute** that partitions the population into two groups. From a statistical perspective, we assume that each sample $(x^{(i)}, a^{(i)}, y^{(i)})$ is a realization of the joint distribution of $P(X, A, Y)$. We denote random variables by capitals, write $p$ for their density, and $P$ for their distributions.

A **subgroup** selects a subset of individuals with shared feature characteristics through its membership function $s : \mathcal{X} \to \{0, 1\}$. The subgroup membership $s(X) = 1$ indicates that an individual with features $X$ belongs to the subgroup, while $s(X) = 0$ indicates non-membership. We denote the distribution of the target variable within the subgroup, for group $a \in \{0, 1\}$ as

$$P_{a,s}(Y) \coloneqq P(Y \mid A = a, s(X) = 1) \,. \tag{1}$$

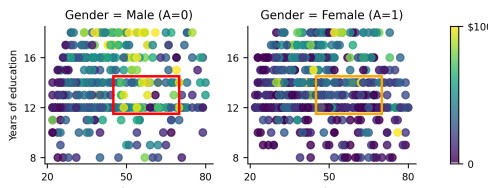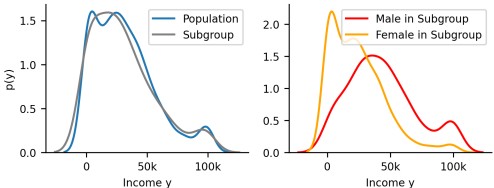

(a) Wage distribution for men (left) and women (right) in relation to age and education level.

(b) Wage distribution of population vs subgroup (left) and of men vs women in subgroup (right).

Figure 1: The income distribution of the subgroup $s$ with `Age` $\in [45, 70]$ and `Education` $\in [12, 14]$ does not deviate significantly from the overall population (left). But, when comparing men and women within that subgroup, there is a significant difference comparing the respective distributions (right). This subgroup is contrastively exceptional.

## 4 CONTRASTIVE SUBGROUPS

From the space of all possible subgroups, our goal is to discover a subgroup $s \in \mathcal{S}$, for which the group membership $A \in \{0, 1\}$ results an exceptional difference in the distribution of the target variable $P_{0,s}(Y)$ and $P_{1,s}(Y)$. In the following, we motivate and formally define the properties of an interesting contrastive subgroup $s$.

**Exceptionality** Unlike regular subgroup discovery, where the target distribution within the subgroup is compared to the overall distribution, we are concerned with the contrast between two given populations. Our objective is to find a local subregion of the feature space, where the difference between the two populations is exceptional. That is, we contrast the subgroup-conditioned target distribution of the first population, $P_{0,s}(Y)$, with the subgroup-conditioned target distribution of the other population, $P_{1,s}(Y)$, using a divergence measure $D$.

**Definition 1 (Contrastive Exceptionality)** *The contrastive exceptionality of a subgroup $s$ is defined as the divergence $D$ between the target variable distributions in the two groups:*

$$\mathcal{E}(s) = D\left(P_{0,s}(Y), P_{1,s}(Y)\right) . \tag{2}$$

We show the difference between contrastive and non-contrastive exceptionality in Figure 1. Under the same subgroup $s$, the overall distribution $P(Y)$ and the subgroup-conditioned distribution $P(Y \mid s(X) = 1)$ are virtually identical. However, if we contrast the two populations, we observe a significant difference in the income of men and women in this particular subgroup. That is, contrastive exceptionality focuses only on the difference *within* the subgroup, independent of the distribution outside the subgroup. This enables it to discover characteristics which one can not recover using regular subgroup discovery methods.

The choice of divergence measure $D$ is flexible in our framework. It can be any measure that quantifies the difference between two distributions, e.g. the Kullback-Leibler/Jensen-Shannon divergence, or Wasserstein distance. In general, the higher the divergence, the more exceptional the subgroup is.

**Support** To be able to compare the target distribution between the two populations, any subgroup $s$ must have support in both populations, i.e. $E[s(X) \mid A = 0] > 0$ and $E[s(X) \mid A = 1] > 0$. On a specific dataset $\{x^{(i)}\}_{i=1}^{n}$, this means that there must be at least one sample $x^{(i)}$ from each group where $s(x^{(i)}) = 1$. In practice, supports beyond the minimal requirement are desirable, such that a subgroup $s$ is supported by a significant portion of the population. Large supports also yield more reliable estimates of $P_{0,s}(Y)$ and $P_{1,s}(Y)$.

**Definition 2 (Generality)** *We define the generality of a subgroup $s$ over two populations as the geometric mean over the individual supports as per*

$$\mathcal{G}_\gamma(s) = \left(\sqrt{E[s(X) \mid A = 0] \cdot E[s(X) \mid A = 1]}\right)^{\gamma} . \tag{3}$$

Under the generality $\mathcal{G}_\gamma$, a subgroup $s$ is supported, iff $\mathcal{G}_\gamma(s) > 0$ which is the case if both $E[s(X) \mid A = 0] > 0$ and $E[s(X) \mid A = 1] > 0$. The parameter $\gamma$ controls the sensitivity of the generality to the individual supports. Larger values of $\gamma$ penalize subgroups with small supports more heavily, while smaller values allow more flexibility in subgroup size.

**Sufficiency**  The trade-off between exceptionality against generality is the standard objective in subgroup discovery. However, in the contrastive setting, we also want to be certain that the observed difference is driven by the subgroup membership rather than varying features within the subgroup. By modeling the unconditional distribution within each subgroup, we implicitly assume that the subgroup membership captures the relevant information about the target variable. To that end, we define the third property relevant to contrastive subgroups.

**Definition 3 (Covariate Dependence)** *We define the covariate dependence as:*

$$\mathcal{C}_0(s) = E_{X \mid A=0, s(X)=1} \left[ D \left( P_{0,s}(Y \mid X = x), P_{0,s}(Y) \right) \right] \tag{4}$$

*and $\mathcal{C}_1(s)$ analogously for the second group. The covariate dependence $\mathcal{C}(s)$ is then given by*

$$\mathcal{C}(s) = \mathcal{C}_0(s) + \mathcal{C}_1(s) . \tag{5}$$

The covariate dependence $\mathcal{C}(s)$ measures how much the target distribution within the subgroup is influenced by the features. In the ideal case, the subgroup membership $s$ captures all relevant information about the target variable $Y$ and $\mathcal{C}(s) = 0$. Then the local target is conditionally independent of the features $X$, i.e. $(Y \perp X) \mid (s(X) = 1, A = 0)$ and $(Y \perp X) \mid (s(X) = 1, A = 1)$. That means that the observed difference in the target variable can solely be attributed to the subgroup membership $A$ alone, and not due to any confounding effect from the features $X$.

Minimizing the covariate dependence $\mathcal{C}(s)$ aims to minimize the influence of indirect causal effects and thus avoid discovering false positive subgroups. By ensuring that the features do not locally influence the target variable, we can be more confident that the observed differences are not due to a shift in features caused by $A$, but instead directly driven by $A$ itself. We later show that under certain conditions, if the covariate dependence is minimized, i.e. $\mathcal{C}(s) = 0$, then the subgroup $s$ captures the interventional distribution of the target variable within that subgroup.

### 4.1 PROBLEM STATEMENT

We can now define the problem of discovering contrastive subgroups as follows:

$$\arg\max_{s \in \mathcal{S}} \mathcal{G}_\gamma(s) \cdot \mathcal{E}(s) - \lambda \mathcal{C}(s) , \tag{6}$$

where $\lambda > 0$ is a hyperparameter that controls the trade-off between exceptionality and covariate independence. The objective function captures the desire to find subgroups that are both general and exceptional, while also ensuring that the observed differences in the target variable are not driven by the features within the subgroup. Depending on the causal relationships between $X$, $A$, and $Y$, the discovered subgroups have different interpretations.

## 5 CAUSALITY OF CONTRASTIVE SUBGROUPS

In the previous section, we formalized the notion of exceptionality for contrastive subgroups. Beyond a high exceptionality score, we also want to ensure that the discovered subgroups are actionable. In this section, we will analyze the causal nature of contrastive subgroups and under which conditions they can be used to make causal inferences. We will use Pearl's $do$-calculus (Pearl, 2009) to analyze the causal relationships between the group membership $A$, features $X$, and target $Y$.

### 5.1 OBSERVATIONAL TRIAL ($X \to A, X \to Y, A \to Y$)

In an observational trial, the goal is to study the effect between treatment and control group, where membership $A$ is not randomized, but rather determined by the features $X$ (Fig. 2a). Due to confounding by $X$, the observational distribution $P(Y \mid A = a)$ differs from the interventional $P(Y \mid do(A = a))$. That is, $X$ influences both the treatment assignment $A$ and the target $Y$.

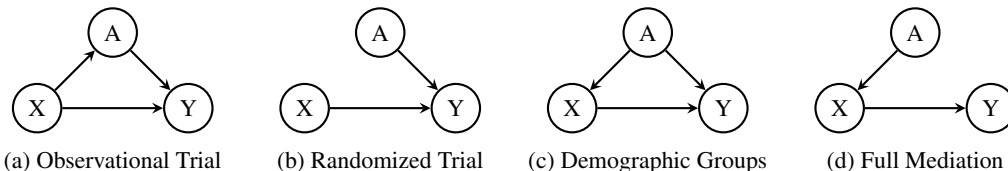

Figure 2: Structural causal models for group indicator $A$, features $X$, and target $Y$. When the covariate dependence is minimized, i.e. $\mathcal{C}(s) = 0$, contrastive subgroups capture the interventional distribution (**a**) or the controlled direct effect (**c**) of $A$ on $Y$.

Under certain conditions however, it is possible to identify the interventional distribution. If $X$ satisfies the backdoor criterion, i.e. if $X$ blocks all backdoor paths between $A$ and $Y$, then the interventional distribution can be identified as

$$p(y \mid do(A = a)) = \int_x p(y \mid X = x, A = a)p(X = x) . \tag{7}$$

That is, if $X$ is a valid backdoor, then it is possible to infer the interventional distribution by integrating over the features $X$. On the other hand, we condition on groups of datapoints for which $s(X) = 1$. This distribution is not necessarily equal to the interventional distribution, due to the influence of the features $X$. However, if the covariate dependence is minimized, i.e. $\mathcal{C}(s) = 0$, then the subgroup $s$ captures the interventional distribution within that subgroup.

**Proposition 1** *Let $X$ be a valid backdoor to infer the interventional distribution as per Eq. (7). Then, any subgroup $s$ that minimizes the covariate dependence $\mathcal{C}(s) = 0$ with generality $\mathcal{G}_\gamma(s) > 0$ captures the interventional distribution within the subgroup*

$$P_{a,s}(Y) = P(Y \mid do(A = a), s(X) = 1) . \tag{8}$$

We provide the proof in Appendix A.2. The result suggest that given data from an observational trial, contrastive subgroups search for a set of data points with $s(X) = 1$ that capture the interventional distribution of the target variable $Y$, and which have an exceptional treatment effect.

## 5.2 Randomized Trial ($X \to Y, A \to Y, A \perp X$)

If the treatment assignment $A$ is randomized, i.e. the treatment assignment $A$ is independent of the covariates $X$, the interventional distribution is equal to the observational distribution. This extends naturally to all sub-distribution induced by a subgroup $s$, regardless of whether the covariate dependence is minimal or not, i.e. $\mathcal{C}(s) > 0$.

Still, it can be of interest to minimize $\mathcal{C}(s)$ alongside. A low covariate dependence mainly ensures that the difference in target distribution is driven by $A$ and not some spurious dependence on $X$. In general, in a randomized control trial, contrastive subgroups may inform treatment policies by discovering patients with exceptional treatment effects.

## 5.3 Demographic Groups ($A \to X, X \to Y, A \to Y$)

Next, we examine the setting where the attribute $A$ has an influence on the feature distribution $P(X)$. This structure is common when searching for disparities between protected demographic groups, where membership $A$ is an inherent attribute like gender or ethnicity that can influence mediating features $X$ (e.g., job role, income level).

The effect of $A$ on $Y$ is twofold: Firstly, $A$ can have a direct effect ($A \to Y$). Secondly, it can have an indirect effect mediated through the features ($A \to X \to Y$).

To distinguish these pathways, causal inference defines the Controlled Direct Effect (CDE). The CDE measures the effect of changing $A$ while holding the mediating features $X$ constant at a specific level $x$. Following Pearl (2022), it is defined as:

$$\text{CDE}(x) = E[Y \mid do(A = 1), do(X = x)] - E[Y \mid do(A = 0), do(X = x)] . \tag{9}$$

Our framework aims to find subgroups where this causal relationship is simplified. If a subgroup $s$ satisfies the conditional independence $(Y \perp X) \mid (A = a, s(X) = 1)$ in the case of $\mathcal{C}(s) = 0$, then the indirect path through $X$ is effectively neutralized within that subgroup. For such a subgroup, the CDE becomes constant for all members and is identifiable directly from the observed difference in means for any member $x$ with $s(x) = 1$:

$$\mathrm{CDE}(x) = E[Y \mid A = 1, s(X) = 1] - E[Y \mid A = 0, s(X) = 1] \,. \tag{10}$$

For mediating variables, our objective trades off discovering subgroups with a large total effect, no matter direct or indirect, against subgroups with a large direct effect, when increasing the regularization parameter $\lambda$.

### 5.4 Full Mediation ($A \to X \to Y$ and $(Y \perp A) \mid X$)

Finally, we consider a scenario of full mediation, where there is no direct causal effect of $A$ on $Y$. Often times, it is unclear whether a direct effect exists or not. In this case, we can use contrastive subgroups to disprove the absence of a direct effect. The following proposition shows that if the model of full mediation is correct, it is impossible to find a subgroup that both has no covariate dependence and exhibits an exceptional outcome. Therefore, if we do find such a subgroup in practice, it serves as evidence against the full mediation assumption, suggesting a direct effect likely exists. We provide the complete proof in the Appendix A.3.

**Proposition 2** *In a model of full mediation, where $A \to X \to Y$ and $(Y \perp A) \mid X$, and the supports of $x$ within each population are equivalent, i.e. $p(x \mid A = 0) > 0 \iff p(x \mid A = 1) > 0$, there cannot exist a subgroup $s$ that simultaneously satisfies $\mathcal{C}(s) = 0$ and $\mathcal{E}(s) > 0$.*

### 5.5 General Discussion

This section has established a causal foundation for contrastive subgroups, demonstrating that their interpretation is fundamentally tied to the assumed causal structure of the data. We have shown that under different, plausible causal models, the discovery of an exceptional subgroup has a distinct meaning. In observational studies, we can identify the local average treatment effect, while in settings with demographic attributes, we isolate the controlled direct effect. This transforms subgroup discovery from a purely descriptive pattern-finding tool into a more powerful instrument.

## 6 Method

In this section, we introduce SUBCON, a practical way to discover contrastive subgroups given a dataset of $n$ individuals $(x^{(i)}, a^{(i)}, y^{(i)})$. We focus on tabular data with a continuous feature space $X \in \mathbb{R}^d$, where we one-hot encode the categorical features. Regarding the target variable, we present solutions for a discrete-valued as well as an uni-variate continuous target $Y$.

### 6.1 Differentiable Rule Learning

To capture the characteristics of a subgroup, we use a rule based membership function $s : \mathbb{R}^d \to [0, 1]$. In this work, we focus on logical conjunctions over single-feature conditions, as used by decision trees and regular subgroup discovery methods (Atzmueller, 2015).

We encode the presence of a characteristic in a feature $j \in [d]$ as $\pi(x_j; a_j, b_j) = \mathbb{1}(a_j < x_j < b_j)$. The rule function $s$ aggregates the conditions over all dimensions with parameters $\theta = \{a_j, b_j\}_{j=1}^d$ using a logical conjunction

$$s(x; \theta) = \bigwedge_{j=1}^d \pi(x_j; a_j, b_j) \,. \tag{11}$$

We instantiate SUBCON using the differentiable rule learner $\hat{s}_t : \mathbb{R}^d \to [0, 1]$ from the SYFLOW framework for subgroup discovery (Xu et al., 2024). The parameters of $\hat{s}_t$ are learned by gradient descent and hence do not require restrictive pre-processing and scale well to high-dimensional data.

We briefly summarize the key components of the differentiable rule learner $\hat{s}_t$. For each feature $j$, the learner places soft conditions $\hat{\pi}_t(x_j; a_j, b_j)$ with learnable thresholds $a_j, b_j \in \mathbb{R}$ as per

$$\hat{\pi}_t(x_j; a_j, b_j) = \frac{1}{1 + \exp\left(-\frac{x_j - a_j}{t}\right) + \exp\left(-\frac{b_j - x_j}{t}\right)} \,, \tag{12}$$

where $t$ is a temperature parameter that controls the smoothness. When annealing the temperature $t \to 0$, the condition approaches the binary thresholding function (Xu et al., 2024). To combine the conditions into a logical conjunction, the weighted harmonic mean is used as per

$$\hat{s}_t(x; \theta, \mathbf{w}) = \frac{\sum_{j=1}^d w_j}{\sum_{j=1}^d w_j \cdot \hat{\pi}(x_j; a_j, b_j, t)^{-1}} \quad \text{with} \quad w_j \in \mathbb{R}_0^+ \,. \tag{13}$$

The weights $w_j$ allow to exclude conditions from the rule by setting $w_j = 0$. Over the remaining conditions, the soft rule $\hat{s}_t(x; \theta, \mathbf{w}) = 1$ only if all conditions $\hat{\pi}_t(x_l; a_l, b_l) = 1$. On the other hand, if there is a condition which is not met with $w_l > 0$, the soft rule tends to zero. Hence, for the binary conditions obtained in the limit of $t \to 0$, $\hat{s}$ mimics the behavior of a logical conjunction, whilst flexibly learning which features to include ($w_j$) and where to place the thresholds ($a_j, b_j$).

## 6.2 Optimization Objective

Using the differentiable rule learner $\hat{s}_t$, we now instantiate the objective proposed in Equation (6). For brevity, we will omit the dependence on the parameters $\theta$ and $\mathbf{w}$ in the following.

**Generality.** We interpret the output of the soft rule $\hat{s}_t : \mathbb{R}^d \to [0, 1]$ as the probability that a sample $x$ belongs to the subgroup $s$, i.e. $\hat{s}_t(x) = \hat{\Pr}(s(x) = 1)$. Over a dataset $\{x^{(i)}, a^{(i)}\}_{i=1}^n$, let $n_0 = \sum_{i=1}^n \mathbb{1}(a^{(i)} = 0)$ and $n_1 = \sum_{i=1}^n \mathbb{1}(a^{(i)} = 1)$. Then the generality of $\hat{s}_t$ is computed as

$$\mathcal{G}_\gamma(\hat{s}_t) = \left( \frac{1}{n_0} \sum_{\{i:a^{(i)}=0\}} \hat{s}_t(x^{(i)}) \cdot \frac{1}{n_1} \sum_{\{i:a^{(i)}=1\}} \hat{s}_t(x^{(i)}) \right)^{\gamma/2} . \tag{14}$$

**Exceptionality.** To compute the exceptionality, we need to estimate the target variable distributions $\hat{P}_{0,s}(Y)$ and $\hat{P}_{1,s}(Y)$. We weight each sample $y^{(i)}$ by the subgroup membership $\hat{s}_t(x^{(i)})$ and use for a *discrete* target $Y \in \{0, \ldots, m\}$ the empirical distribution, and a kernel density estimator for a *continuous* target $Y \in \mathbb{R}$. During training, we refit the estimators after every $k = 10$ updates to $\hat{s}_t$.

Now that we have the target distributions, we can compute the contrastive exceptionality (Def. 1). As divergence measure $D$, we use the Jensen–Shannon divergence,

$$D_{\text{JS}}(P, Q) = \frac{1}{2} D_{\text{KL}}(P, M) + \frac{1}{2} D_{\text{KL}}(Q, M), \quad M = \tfrac{1}{2}(P + Q).$$

Compared to the Kullback–Leibler divergence, $D_{\text{JS}}$ seamlessly handles cases where the supports of $P$ and $Q$ do not overlap. Let the mixture density be $m(y) = \frac{1}{2}(\hat{p}_{0,s}(y) + \hat{p}_{1,s}(y))$. Using the approximation of the KL divergence under subgroup membership (Xu et al., 2024),

$$D_{\text{KL}}\left(\hat{P}_{a,s}(Y), M(Y)\right) \approx \frac{1}{n_a} \sum_{\{i:a^{(i)}=a\}} \hat{s}_t(x^{(i)}) \log\left(\frac{\hat{p}_{a,s}(y^{(i)})}{m(y^{(i)})}\right), \tag{15}$$

we obtain the final objective

$$\mathcal{E}(\hat{s}_t) = \frac{1}{2} D_{\text{KL}}(\hat{P}_{0,s}(Y), Q(Y)) + \frac{1}{2} D_{\text{KL}}(\hat{P}_{1,s}(Y), Q(Y)). \tag{16}$$

**Covariate Dependence.** Lastly, the covariate dependence $\mathcal{C}(s)$ is needed to regulate the influence of the features $X$ on the target variable $Y$. To that end, we also need to estimate $\hat{P}(Y \mid X = x, A = a)$. For a *discrete* target variable, we train a random forest model $f(x)$ to predict the class probabilities $\hat{\mathbb{P}}(Y = l \mid X = x, A = a)$. For each sample $x^{(i)}$, we compute a local divergence as

$$c(x^{(i)}, a^{(i)}) = \sum_{l=1}^m \hat{\mathbb{P}}(Y = l \mid X = x^{(i)}, A = a^{(i)}) \log\left(\frac{\hat{\mathbb{P}}(Y = l \mid X = x^{(i)}, A = a^{(i)})}{\hat{\mathbb{P}}_{a^{(i)},s}(Y = l)}\right) . \tag{17}$$

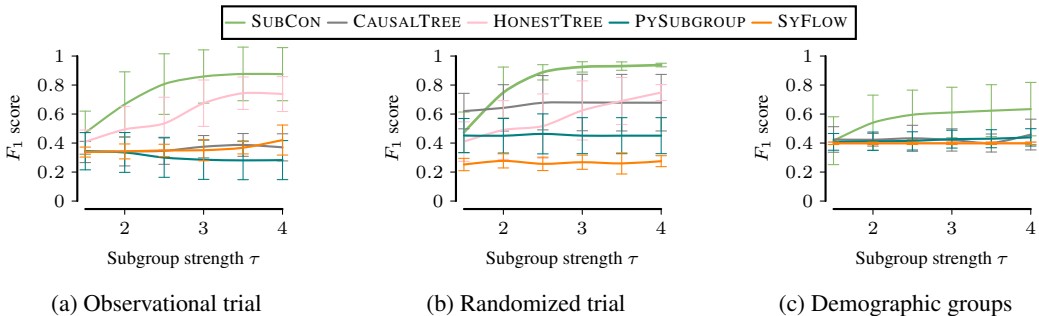

(a) Observational trial  (b) Randomized trial  (c) Demographic groups

Figure 3: Performance of each method in the synthetic experiments. SUBCON objective of discovering contrastive subgroups allows it to outperform all other methods in all settings.

For a *continuous* target variable, assessing the conditional distribution is more difficult. Therefore, to bypass the need for a full conditional distribution, we compare only the first moment of the local resp. subgroup distribution. For a continuous target variable $Y$, we likewise fit a random forest to compute the squared difference between the local mean and the subgroup mean as

$$c(x^{(i)}, a^{(i)}) = (f(x^{(i)}, a^{(i)}) - E[Y \mid A = a^{(i)}, S = 1])^2 \ . \tag{18}$$

Then, the covariate dependence is computed as

$$\mathcal{C}(\hat{s}_t) = \sum_{i:a^{(i)}=0} \hat{s}_t(x^{(i)}) \cdot c(x^{(i)}, a^{(i)}) + \sum_{j:a^{(j)}=1} \hat{s}_t(x^{(j)}) \cdot c(x^{(j)}, a^{(j)}) \tag{19}$$

Thus, we have obtained a fully differentiable approach to discovering contrastive subgroups. We describe the full training procedure in detail in Appendix B.

## 7 EXPERIMENTS

We compare SUBCON against regular subgroup discovery using the PYSUBGROUP package (Lemmerich & Becker, 2018) as well as SYFLOW (Xu et al., 2024) using the authors implementation. We also compare against CAUSALTREE, which implement recursive partitioning for randomized trials using the causalml package (Chen et al., 2020), and HONESTTREE (Athey & Imbens, 2016), which focuses on observational trials using econml (Keith Battocchi, 2019).

### 7.1 SIMULATED DATA

We first evaluate on simulated data with known causal structure, following the observational, randomized and demographic setting from Figure 5. The target is defined as $Y = f(X, A) + N_Y$ with Gaussian noise, where $X$ may be a cause or effect of $A$. We plant a subgroup $s$ using two random features, scaling the treatment effect by $\tau$ within the subgroup: $f(x, 1) = f(x, 0) + \tau$ if $s(x) = 1$, else $f(x, 1) = f(x, 0) + 1$ (see Appendix C.1). Hyperparameters are tuned via grid search (Appendix D). Performance is measured by $F_1$-score between the true and discovered subgroup, where we report the best-scoring leaf for tree-based methods.

Figure 3 reports $F_1$-scores as $\tau$ increases. PYSUBGROUP and SYFLOW fail due to their non-contrastive objectives. CAUSALTREE excels in randomized trials but struggles in observational ones, while HONESTTREE shows the opposite pattern. Both degrade in the demographic setting due to indirect effects. SUBCON consistently recovers the correct subgroup, improving with larger $\tau$. By targeting contrastive subgroups, SUBCON outperforms specialized baselines across all settings.

We further conduct a sensitivity analysis on the parameters $\lambda$ and $\gamma$ as well as the choice of estimator in Appendix D.1. The results for $\gamma$ show that performance is stable in a range of $[0.1, 0.3]$, while large values or disabling the generality ($\gamma = 0$) degrade performance. For $\lambda$, we find that the performance is stable up to a $\lambda = 0.5$ and then degrades for larger values. All trends persist across the different settings, showing that they are not specific to one causal structure.

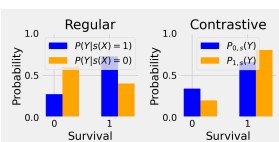
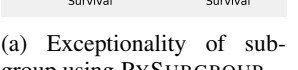

(a) Exceptionality of subgroup using PYSUBGROUP

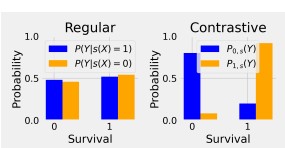

(b) Exceptionality of subgroup using SUBCON

Figure 4: Subgroup discovery on COVID-19 dataset.

|  | IHDP | |
| Method | Subgroup | Overall |
| --- | --- | --- |
| SUBCON | 1.73 | 4.00 |
| Ablation | 3.77 | 4.39 |
| HONESTTREE | 2.33 | 3.93 |
| CAUSALFOREST | 2.17 | 5.52 |
| XLEARNER | 1.19 | 2.07 |

Table 1: Precision in estimating heterogeneous effects on IHDP.

## 7.2 REAL-WORLD DATA

We evaluate SUBCON on the IHDP benchmark for conditional average treatment effect (CATE) estimation (Hill, 2011), using 10 semi-synthetic datasets from Louizos et al. (2017). Baselines include HONESTTREE (Athey & Imbens, 2016), CAUSALFOREST (Wager & Athey, 2018), and the non-interpretable XLEARNER (Künzel et al., 2019) with Gradient Boosted Trees. CATE for SUBCON is estimated as the mean difference between treated and control units inside vs. outside the discovered subgroup. We also test an ablation with $\lambda = 0$ to assess the efficacy of our regularization.

Table 1 reports the precision in estimating heterogeneous effects (PEHE) both within the discovered subgroup and over the full dataset. Within the subgroup, SUBCON achieves a lower PEHE (1.73) compared to HONESTTREE (2.33) and CAUSALFOREST (2.17). The ablation performs markedly worse (3.77), confirming that regularization helps identify subgroups that accurately reflect treatment effects (Prop. 1). Outside the subgroup, however, SUBCON 's overall PEHE (4.00) is worse than HONESTTREE (3.93) and far inferior to XLEARNER (2.07). This shows that SUBCON is well-suited for detecting CATE subgroups, but not for modeling the entire population.

Lastly, we compare contrastive vs regular subgroup discovery on the COVID-19 dataset from Lambert et al. (2022), containing biomarkers and outcomes of ICU patients from two New York City hospitals. The target is binary mortality, with features including demographics, comorbidities, and vital signs. We split by gender and use SUBCON to identify subgroups with differential mortality between men and women, reporting the top subgroup found by PYSUBGROUP: " Age $<$ 74 & Coronary Artery $= 0$ & Cerebrovascular $= 0$" and by SUBCON: " Race $=$ black & Diabetes $= 1$ & Cerebrovascular $= 0$ & Hypertension $= 0$".

We visualize these results in Figure 4. The left panels show mortality inside vs. outside the subgroup (standard subgroup discovery), and the right panels compare mortality *within* the subgroup between men and women. PYSUBGROUP (Fig. 4a) identifies younger patients without severe comorbidities, who overall have higher mortality (Szarpak et al., 2022), but show little gender difference. In contrast, SUBCON (Fig. 4b) identifies black patients with diabetes, for whom mortality is markedly higher in men than women—though not elevated relative to the overall population. Such a pattern is only detectable with a contrastive objective.

## 8 CONCLUSION

We introduce contrastive subgroups that describe subsets showing exceptional disparities in a target variable. They are defined by three key properties: support in both populations, a significant outcome difference, and minimal confounding. Our method, SUBCON, discovers such subgroups via a continuous objective that captures these criteria. It enables causal insights in both observational and randomized trials and reveals disparities in demographic groups. In experiments, SUBCON outperforms existing methods on synthetic data and yields meaningful subgroups in real-world datasets.

**Limitations.** SUBCON assumes that observed features $X$ capture relevant causal relationships. Unobserved confounding or post-treatment variables may compromise causal validity, so domain-informed variable selection is crucial. Regularization relies on accurate estimation of $P(Y \mid X, A)$; poor estimates can affect subgroup quality. Currently, our framework identifies only a single contrastive subgroup; to discover multiple subgroups we currently use sequential re-learning with exclusion of previously found subgroups. Future work will explore more principled approaches.

## ETHICS STATEMENT

SUBCON is designed to support practitioners in identifying interesting subgroup differences. When applied to the medical or social science domain for example, the discovered subgroups can highlight disparities or treatment heterogeneity with potential real world impact. However, we emphasize that the discovered subgroups lend themselves to causal conclusions only if the stated assumptions are satisfied. In sensitive applications, such as fairness analysis, healthcare, or policy, results should therefore be interpreted with caution, always in consideration of existing domain knowledge. Contrastive subgroup discovery is to be used for hypothesis generation and exploratory data analysis, which are further validated, rather than to draw definite conclusions that may reinforce biases.

## REPRODUCIBILITY STATEMENT

We provide the code to reproduce all experiments in the supplementary material, including a requirements file with all necessary dependencies. This includes the implementation of SUBCON, the baselines, and the datasets. We also provide detailed instructions to run the experiments, including the hyperparameter settings.

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

# A  PROOFS

In this section, we provide a thorough investigation of the formal and causal properties of contrastive subgroups.

## A.1  COVARIATE DEPENDENCE

We first show the connection between a minimized covariate dependence $\mathcal{C}(s) = 0$ (Def. 3) and the independence of features.

**Lemma 1** *Given a distribution distance $D : \mathcal{P}(\mathcal{Y}) \times \mathcal{P}(\mathcal{Y}) \rightarrow [0, \infty)$, where it holds that $D(P, Q) = 0$ if and only if $P = Q$. A minimal covariate dependence implies a conditional independence between the features $X$ and the target $Y$ within the scope of the subgroup $s(X)$, i.e.*

$$\mathcal{C}(s) = 0 \Rightarrow (Y \perp X) \mid (A, s(X) = 1) .$$

**Proof:**  *The covariate dependence is defined as*

$$\mathcal{C}_0(s) = E_{X|A=0,s(X)=1} \left[ D \left( P_{0,s}(Y \mid X = x), P_{0,s}(Y) \right) \right] ,$$

*for $A = 0$ and analogously for $A = 1$. It is comprised of the integral over the feature space $\mathcal{X}$ as*

$$\mathcal{C}_0(s) = \int_x D \left( P_{0,s}(Y \mid X = x), P_{0,s}(Y) \right) p(x \mid A = 0, s(X) = 1) dx .$$

*Given that $\mathcal{C}_0(s) = 0$ and $D$ has a domain of $[0, \infty)$, it holds that for every $x \in \mathcal{X}$ with $p(x \mid A = 0, s(X) = 1) > 0$ the conditional and target distribution are equal, i.e.*

$$\forall x \in \mathcal{X}, p(x \mid A = 0, s(X) = 1) > 0 : P(Y \mid A = 0, X = x, s(X) = 1) = P(Y \mid A = 0, s(X) = 1) .$$

*Therefore, it holds that*

$$P(Y \mid X = x, A = 0, s(X) = 1) = P(Y \mid A = 0, s(X) = 1) ,$$

*i.e. $(Y \perp X) \mid (A = 0, s(X) = 1)$. We can similarly derive the same result for $A = 1$. This shows the claim that if $\mathcal{C}(s) = 0$, the features $X$ and $Y$ are conditionally independent given $s(X) = 1$ and $A$.* □

## A.2  PROOF OF PROPOSITION 1

**Proposition 1** *Let $X$ be a valid backdoor to infer the interventional distribution as per Eq. (7). Then, any subgroup $s$ that minimizes the covariate dependence $\mathcal{C}(s) = 0$ with generality $\mathcal{G}_\gamma(s) > 0$ captures the interventional distribution within the subgroup*

$$P_{a,s}(Y) = P(Y \mid do(A = a), s(X) = 1) . \tag{8}$$

**Proof:**  *Firstly, the population support $\mathcal{G}_\gamma(s) > 0$ ensures positivity, i.e. that there are samples from both populations within the subgroup, i.e. $P(s(X) = 1 \mid A = a) > 0$ for $a \in \{0, 1\}$.*

*We start with the **interventional distribution** the right-hand side. Since $X$ is a valid backdoor, we can identify the causal effect within the subgroup by adjusting for $X$:*

$$P(Y \mid do(A = a), s(X) = 1) = \int_x P(Y \mid A = a, X = x, s(X) = 1) p(x \mid s(X) = 1) dx .$$

*If the covariate dependence is minimized so that $\mathcal{C}(s) = 0$ and $D$ is a distribution distance for which $D(P, Q) = 0$ iff $P = Q$, then as per Lemma 1 we know that $(Y \perp X) \mid (A, s(X) = 1)$.*

*This means that $P(Y \mid A = a, X = x, s(X) = 1) = P(Y \mid A = a, s(X) = 1)$. Substituting this into the equation yields*

$$P(Y \mid do(A = a), s(X) = 1) = \int_x P(Y \mid A = a, s(X) = 1) p(x \mid s(X) = 1) dx$$

$$= P(Y \mid A = a, s(X) = 1) \int_x p(x \mid s(X) = 1) dx$$

The summation term $\int_x p(x \mid s(X) = 1)$ is the integrated density of $X$ over its support within the subgroup, which equals 1.

$$P(Y \mid do(A = a), s(X) = 1) = P(Y \mid A = a, s(X) = 1) \cdot 1 = P_{a,s}(Y) \ .$$

This shows that under the condition of zero covariate dependence, the interventional distribution (LHS) within the subgroup simplifies to the observational distribution (RHS), which completes the proof. □

## A.3 PROOF OF PROPOSITION 2

**Proposition 2** *In a model of full mediation, where $A \to X \to Y$ and $(Y \perp A) \mid X$, and the supports of $x$ within each population are equivalent, i.e. $p(x \mid A = 0) > 0 \iff p(x \mid A = 1) > 0$, there cannot exist a subgroup $s$ that simultaneously satisfies $\mathcal{C}(s) = 0$ and $\mathcal{E}(s) > 0$.*

**Proof:** *We will prove this by contradiction. Assume that in a model of full mediation, there **exists** a subgroup $s$ that simultaneously satisfies both conditions:*

1. *Covariate independence: $\mathcal{C}(s) = 0$, which implies $(X \perp Y) \mid (A, s(X) = 1)$.*

2. *Exceptional outcome: $\mathcal{E}(s) > 0$, which implies that the outcome distributions differ, i.e., $P(Y \mid A = 1, s(X) = 1) \neq P(Y \mid A = 0, s(X) = 1)$.*

*Now, we analyze the consequences of the other assumptions. Let x be any feature vector from the support of the subgroup, i.e., $s(x) = 1$. The premise that the supports overlap ensures such an x exists for both populations.*

*From Condition 1 ($\mathcal{C}(s) = 0$), Lemma 1 tells us that the target and features are conditionally independent given the attribute and subgroup membership. This allows us to state that for any x with $s(x) = 1$, the distribution of $Y$ is constant within the subgroup for a given attribute a:*

$$P(Y \mid A = a, s(X) = 1) = P(Y \mid A = a, X = x, s(X) = 1) \quad \text{for } a \in \{0, 1\}$$

*From the problem setup, we assume a model of full mediation, which is formally defined as $(Y \perp A) \mid X$. This means that once the features x are known, the attribute a provides no additional information about the target $Y$. This gives us the following equality: As per the causal structure of the data, $X$ is a mediator between $A$ and $Y$, so that it holds for every sample x:*

$$P(Y \mid X = x, A = a) = P(Y \mid X = x) \ .$$

*We can now combine these two equalities. Beginning with the $A = 1$ population:*

$$P(Y \mid A = 1, s(X) = 1) = P(Y \mid A = 1, X = x, s(X) = 1) \quad = P(Y \mid X = x, s(X) = 1)$$

*We now apply the same transformation to the $A = 0$ population:*

$$P(Y \mid A = 0, s(X) = 1) = P(Y \mid A = 0, X = x, s(X) = 1) \quad = P(Y \mid X = x, s(X) = 1)$$

*By chaining these equalities, we have shown that both population-specific distributions are equal to the same underlying distribution, $P(Y \mid X = x, s(X) = 1)$. Therefore, they must be equal to each other:*

$$P(Y \mid A = 1, s(X) = 1) = P(Y \mid A = 0, s(X) = 1)$$

*This implies that the exceptionality score $\mathcal{E}(s)$, which measures the divergence between these two distributions, must be 0. This is a direct contradiction to our initial assumption that $\mathcal{E}(s) > 0$. In a model of full mediation, there cannot exist a subgroup $s$ that satisfies both $\mathcal{C}(s) = 0$ and $\mathcal{E}(s) > 0$.* □

## B    Training Details

We implement our method using `PyTorch` (Paszke et al., 2019). We optimize the parameters $\theta$ of the neural network $\hat{s}_t(x; \theta, \mathbf{w})$ and the attribute weights $\mathbf{w}$ using Adam (Kingma & Ba, 2015). We describe the chosen hyperparameters for the optimization in Appendix D.

We provide the pseudo-code of our training procedure in steps below. In every epoch, we perform the following steps:

1. Compute the soft subgroup membership $\hat{s}_t(x^{(i)}; \theta, \mathbf{w})$ for each sample $x^{(i)}$ using Eq. (13).
2. Estimate $\hat{P}_{a,s}(y^{(i)})$ for every sample $y^{(i)}$ using the respective subgroup population density/probability estimator (see below).
3. Compute the exceptionality $\mathcal{E}(\hat{s}_t)$ using Eq. (16).
4. Compute the generality $\mathcal{G}_\gamma(\hat{s}_t)$ using Eq. (14).
5. Estimate the local conditional distribution $\hat{P}(Y \mid X = x, A = a)$ using a regression or classification model (see below).
6. Compute the covariate dependence $\mathcal{C}(\hat{s}_t)$ using Eq. (19).
7. Compute the overall loss $\mathcal{L}(\theta, \mathbf{w})$ using Eq. (6).
8. Update the parameters $\theta, \mathbf{w}$ using gradient descent.
9. Update the temperature $t$ using a linear decay schedule (see below).
10. Every $k = 10$ epochs, re-estimate the density estimator within the subgroup using the updated soft memberships.

### B.1    Density Estimation

For a **discrete** target $Y \in \{0, \ldots, m\}$, we use the empirical distribution weighted by the subgroup membership, e.g.

$$\hat{P}_{0,\hat{s}}(Y = l) = \frac{\sum_{\{i:a^{(i)}=0\}} \hat{s}_t(x^{(i)}) \, \mathbb{1}(y^{(i)} = l)}{\sum_{\{i:a^{(i)}=0\}} \hat{s}_t(x^{(i)})} \, .$$

For a **continuous** target $Y \in \mathbb{R}$, we use a kernel density estimator (Terrell & Scott, 1992) using the implementation from `scipy.stats.gaussian_kde` where each sample is weighted by $\hat{s}_t(x^{(i)})$. The bandwidth is selected using Scott's rule of thumb. This yields two subgroup density estimators $\hat{p}_{0,s}(y)$ and $\hat{p}_{1,s}(y)$.

### B.2    Regularization

As part of the estimation of covariate dependence, we estimate the conditional distribution $\hat{P}(Y \mid X = x)$ for discrete and continuous target variables. For a **discrete** target variable, we train a boosting ensemble $f(x)$ using `RandomForestClassifier` from `sklearn` to predict the class probabilities $\hat{\mathbb{P}}(Y = l \mid X = x)$. For a **continuous** target variable, we train a regression model using `RandomForestRegressor` from `sklearn`. In both models, we use the default hyperparameters of `sklearn` (100 estimators, unrestricted depth, gini criterion and least squares loss respectively). At this time, we do not perform hyperparameter tuning for these models as it would significantly increase the computational cost of training.

### B.3    Temperature Annealing

Temperature schedules are an important mechanism in optimization with soft relaxations of discrete functions. They provide a smooth transition from soft to hard decision boundaries, improving both convergence and final performance. By adjusting the temperature parameter, we control the degree of smoothness: higher values encourage exploration in the early stages of training, while lower values lead to sharper, more precise solutions as training progresses.

We employ a linear decay schedule during the second half of training for the temperature parameter. The temperature $t$ decays from 0.2 to 0.05. These ranges were selected via hyperparameter

optimization and remain fixed across all experiments. The update at each epoch is implemented as follows:

```
temp_start = 0.2
temp_end = 0.05
temp = temp_start
step_size = (temp_start - temp_end)/(total_epochs)
for epoch in range(total_epochs):
    temp = temp - step_size
```

## C  EXPERIMENTS

All experiments were conducted on a consumer grade laptop.

### C.1  SYNTHETIC DATA GENERATION

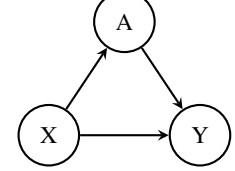
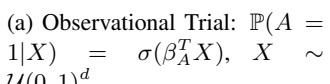
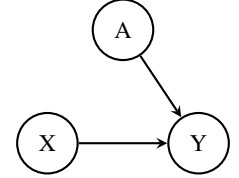
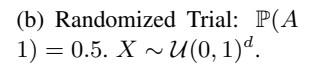
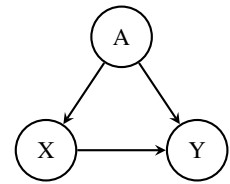

(a) Observational Trial: $\mathbb{P}(A = 1|X) = \sigma(\beta_A^T X)$, $X \sim \mathcal{U}(0,1)^d$.

(b) Randomized Trial: $\mathbb{P}(A = 1) = 0.5$. $X \sim \mathcal{U}(0,1)^d$.

(c) Demographic Groups: $\mathbb{P}(A = 1) = 0.5$. $X \sim \mathcal{U}(0,1)^d + \mathbb{1}(A = 1) \cdot \boldsymbol{\mu}$, where $\boldsymbol{\mu} \in [-0.3, 0.3]^d$.

Figure 5: Data generation process for the three settings: observational trials, randomized trials, and demographic groups. In all settings, the target is a linear function of the covariates with a subgroup-dependent mean-shift between $P(Y|A = 1)$ and $P(Y|A = 0)$ such that $Y = f(X, A) + N_Y$.

To evaluate the performance of methods in recovering contrastive subgroups, we generate datasets under three different causal structures. In all settings, the target variable $Y$ is a function of a binary attribute $A$ and a set of covariates $X$ as per

$$Y = f(X, A) + N_Y ,\tag{20}$$

We summarize the respective data generation process for each setting in Figure 5. The core components of the generated data are:

- **Subgroup** $s^*(X)$: A ground-truth subgroup defined by an axis-aligned box rule on a small subset of features $\mathcal{J} \subset \{1, \ldots, d\}$. A sample $x$ belongs to the subgroup if its covariates $x_j$ satisfy the rule:

$$s^*(x) = \bigwedge_{j \in \mathcal{J}} (a_j < x_j < b_j)$$

  where $a_j$ and $b_j$ are lower and upper bounds for feature $j$.
- **Target Variable** $Y$: A continuous outcome variable that depends on the binary attribute $A$ and a set of covariates $X$ as per

$$Y = f(X, A) + N_Y ,\tag{21}$$

  where $N_Y$ is Gaussian noise with mean zero and variance $\sigma^2$. $f$ consist of a linear function $\beta_Y^T X$, $\beta_Y \in [-1, 1]^d$, with an subgroup dependent interaction term for the difference between $A = 1$ and $A = 0$:

$$f(X, A) = \beta_Y^T X + s^*(X)\left(\frac{\tau}{2}\mathbb{1}(A = 1) - \frac{\tau}{2}\mathbb{1}(A = 0)\right) + (1 - s^*(X))\left(\frac{\eta}{2}\mathbb{1}(A = 1) - \frac{\eta}{2}\mathbb{1}(A = 0)\right) ,\tag{22}$$

  where $\tau$ is the treatment effect for individuals in the subgroup, i.e. $s^*(X) = 1$, and $\eta$ is the treatment effect for individuals outside the subgroup, i.e. $s^*(X) = 0$.

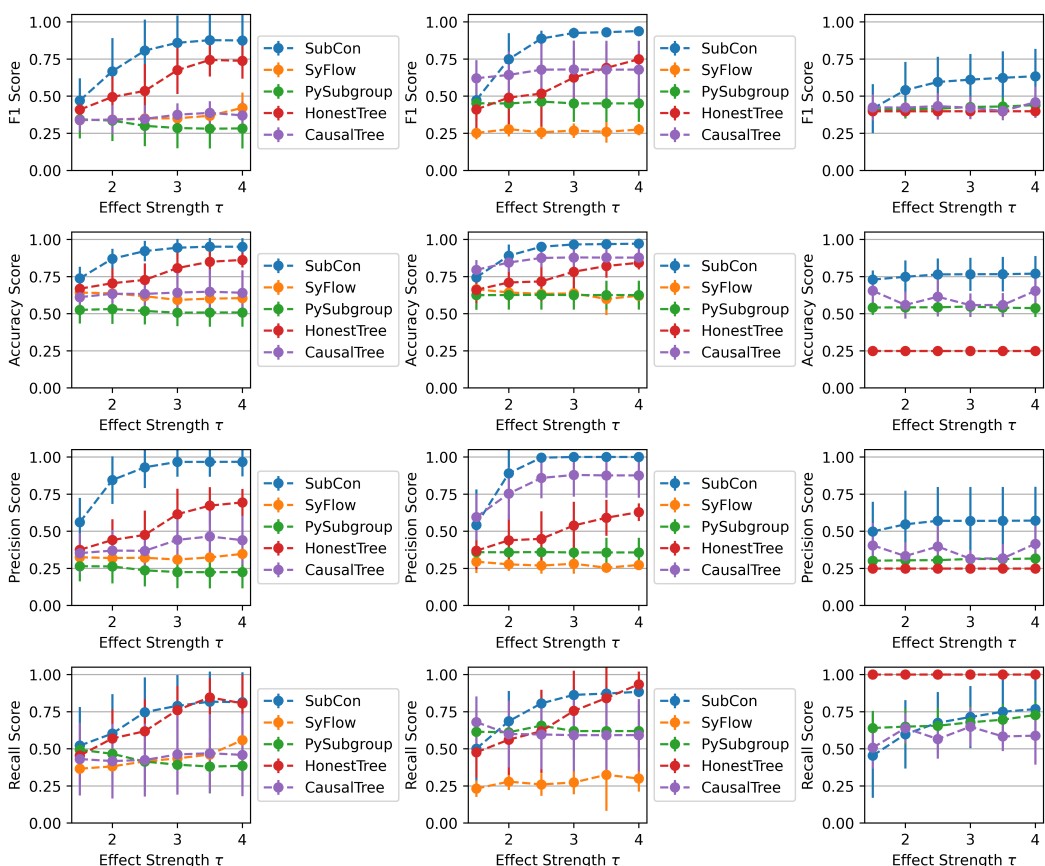

Figure 6: From top to bottom row: $F_1$-score, accuracy, precision, recall of recovered subgroup. SUBCON outperforms the competing methods in the observational (left column), randomized (middle column) and demographic (right column) causal data generating mechanisms.

- **Covariates** $X$: A feature matrix $X \in \mathbb{R}^{n \times d}$. For the setting "observational" and "interventional", each feature $X_j$ is sampled uniformly from the interval $[0, 1]$. In the "demographic shift" setting, the features are mean-shifted conditional on the binary attribute $A$.
- **Population Indicator** $A$: A binary population assignment $A \in \{0, 1\}$. For the setting "interventional" and "demographic" shift, $A$ is sampled uniformly at random, $P(A = 1) = 0.5$. In the "observational" setting, $A$ is sampled from a Bernoulli distribution with probability dependent on the covariates $X$.

Across all settings, we vary the direct effect in the subgroup $\tau \in \{1.5, 2, 2.5, 3, 3.5, 4\}$ compared to a outside effect of $\eta = 1$. The subgroup is defined by two randomly chosen features $X_i$ and $X_j$, with random bounds such that $P(s^*(X) = 1) = 0.3$, i.e. 30% of the samples belong to the subgroup. The number of samples is set to $n = 2000$ and the number of features to $d = 5$. The noise variance is set to $\sigma^2 = 0.5$.

As per the assumed causal struture, we obtain the following distributions for $A$ and $X$:

- **Observational**: The covariates $X$ are sampled uniformly from the interval $[0, 1]^d$. The binary attribute $A$ is sampled from a Bernoulli distribution with probability dependent on the covariates $X$
$$P(A = 1 \mid X) = \sigma(\beta_A^T X) ,$$
where $\sigma(\cdot)$ is the sigmoid function and $\beta_A \in [-1, 1]^d$ is a vector of coefficients that determines the treatment assignment probability.
- **Interventional**: The covariates $X$ are sampled uniformly from the interval $[0, 1]^d$. The binary attribute $A$ is sampled uniformly at random, $P(A = 1) = 0.5$.

- **Demographic Shift**: The binary attribute $A$ is sampled uniformly at random, $P(A = 1) = 0.5$. The covariates $X$ are sampled uniformly from the interval $[0, 1]^d$, but with a mean shift conditional on the binary attribute $A$ as per

$$X_i \sim \mathcal{U}(0, 1)^d + \mu_i \cdot \mathbb{1}(A = 1) \,.$$

The feature shift $\mu_i \in [-0.3, 0.3]$ is the distribution shift of $X_i$ induced by $A$.

### C.2 RESULTS OF SYNTHETIC EXPERIMENTS

We generate data using causal mechanism as detailed above. We evaluate the correctness of the recovered subgroup for each sample $x^{(i)}$, which we denote as $\hat{s}(x_i)$, and compared to the ground truth $s^*(x^{(i)})$.

We report the following metrics :

- $F_1 = \frac{2TP}{2TP+FP+FN}$
- Accuracy $= \frac{TP+TN}{TP+FP+TN+FN}$
- Precision $= \frac{TP}{TP+FP}$
- Recall $= \frac{TP}{TP+FN}$

We display all metrics for the synthetic experiments in Fig. 6. With increasing strength $\tau$ of the direct causal effect, SUBCON recovers the ground truth with high precision and good recall. SUBCON beats the domain specific methods HONESTTREE for observational data and CAUSALTREE for randomized control trials, and is the only one to recover a reasonable result in the demographic setting. On the other hand, the regular subgroup discovery methods PYSUBGROUP and SYFLOW do not work well in any setting, due to their non contrastive approach.

**Runtimes.** We report the runtimes of each method on the synthetic data in Fig. 7. SUBCON is slightly slower than SYFLOW due to the increased complexity of the loss function, but is still in the same order of magnitude. The tree-based methods HONESTTREE and CAUSALTREE are significantly faster. PY-SUBGROUP is faster for few features $d$, but scales poorly with increasing number of features due to its combinatorial search strategy.

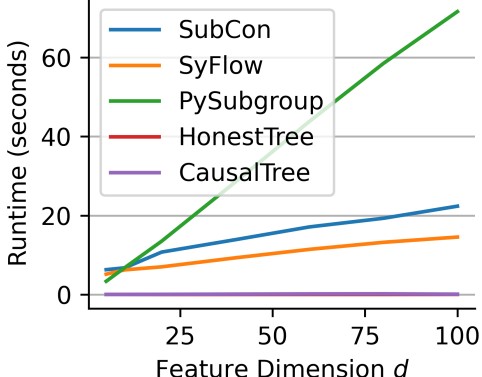

Figure 7: Runtimes of the methods on synthetic data.

### D  HYPERPARAMETER CONFIGURATION

For our experiments, we performed a comprehensive hyperparameter search for each method across the three distinct settings introduced in the synthetic experiments: **observational**, **interventional**, and **demographic**. We employed a grid search, exhaustively testing all combinations of the parameters specified for each algorithm. The rule-based models, SUBCON and SYFLOW, share a common rule configuration where the predicate temperature is initialized at $0.2$ and linearly annealed over the course of training, reducing by a factor of $10$.

We use the setting of $n = 2000$, $d = 5$, $\tau = 4$, $\sigma = 0.5$ for the synthetic data, which is consistent across all methods. We generate 10 different synthetic datasets for each method to ensure robustness in our evaluation and we selected the best-performing configuration for each specific setting based on validation performance.

Table 2: Selected hyperparameters for each method across the different experimental settings.

| Method | Setting | Selected Hyperparameters |
|---|---|---|
| SUBCON | Observational Interventional Demographic | $\lambda = 0.5$, $\gamma = 0.1$, n_epochs=500, lr_classifier=0.005 |
| SYFLOW | Observational | $\alpha = 0.5$, lr_classifier=$10^{-3}$, subgroup_train_epochs=1000 |
| | Interventional | $\alpha = 0.1$, lr_classifier=$10^{-3}$, subgroup_train_epochs=1000 |
| | Demographic | $\alpha = 0.1$, lr_classifier=$10^{-4}$, subgroup_train_epochs=200 |
| PYSUBGROUP | Observational Demographic | beam_width=200, n_bins=20, $\alpha = 1.0$ |
| | Interventional | beam_width=100, n_bins=5, $\alpha = 1.0$ |
| HONESTTREE | Observational | min_samples_leaf=0.01, max_depth=3 |
| | Interventional | min_samples_leaf=0.01, max_depth=5 |
| | Demographic | min_samples_leaf=0.05, max_depth=5 |
| CAUSALTREE | Observational | min_samples_leaf=0.1, max_depth=3 |
| | Interventional | min_samples_leaf=0.1, max_depth=2 |
| | Demographic | min_samples_leaf=0.3, max_depth=5 |

The hyperparameters and their tested ranges for each method are as follows:

- SUBCON: The grid search included the trade-off parameter $\lambda \in \{0.1, 0.5, 1.0\}$, the generality parameter $\gamma \in \{0.1, 0.5\}$, the number of training epochs $\in \{500, 1000\}$, and the classifier's learning rate $\in \{0.001, 0.005\}$.

- SYFLOW: We tuned the size rewards as $\alpha \in \{0.1, 0.3, 0.5\}$, the classifier's learning rate $\in \{10^{-3}, 10^{-4}, 10^{-5}\}$, and the number of subgroup training epochs $\in \{200, 500, 1000\}$.

- PYSUBGROUP: We explored different values for the beam width $\in \{50, 100, 200\}$, the number of bins for discretization $n_{bins} \in \{5, 10, 20\}$, and the size trade-off parameter $\alpha \in \{0.2, 0.5, 1.0\}$.

- HONESTTREE & CAUSALTREE: For both tree-based methods, we searched over the minimum number of samples per leaf, specified as a fraction of the dataset, min_samples_leaf $\in \{0.01, 0.05, 0.1, 0.2, 0.3\}$ and the maximum tree depth max_depth $\in \{2, 3, 5, \text{None}\}$.

We use the validation performance to select the best hyperparameters for each method in each setting. The validation performance is measured by the area under the receiver operating characteristic curve (AUC) for the subgroup prediction task, which is a common metric for evaluating the performance of classification models in subgroup discovery

The table below details the final hyperparameter configurations chosen for each method in the respective experimental settings.

## D.1 SENSITIVITY

Additionally, we conduct a sensitivity analysis of the hyperparameters $\lambda$ and $\gamma$ of SUBCON. We vary $\lambda$ in the range $[0., 1.0]$ and $\gamma$ in the range $[0., 1.0]$. We keep all other hyperparameters fixed to the values selected in the observational setting (see Table 2). The results are displayed in Fig. 8. For $\gamma$, we observe that a small value of $\gamma \in [0.1, 0.3]$ leads to the best performance across all settings. When disabling size regularization, i.e. $\gamma = 0$, the performance drops significantly. On the other hand, when prioritizing size too much, i.e. $\gamma > 0.5$, the performance also drops. For $\lambda$, we observe that a value of $\lambda \in [0.0, 0.5]$ leads to good performance. Increasing the strength

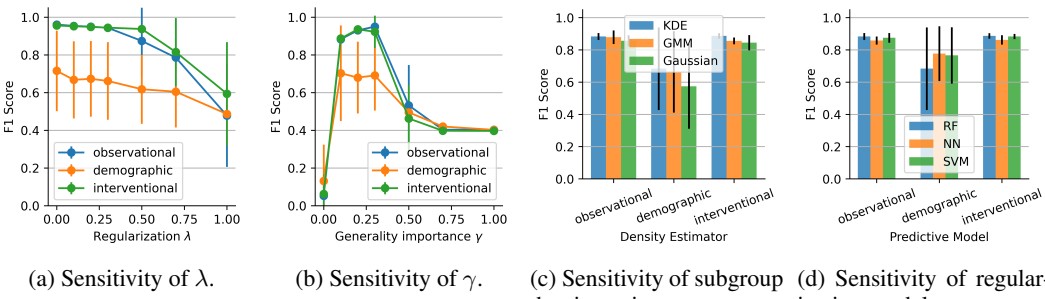

(a) Sensitivity of $\lambda$.     (b) Sensitivity of $\gamma$.     (c) Sensitivity of subgroup density estimator.     (d) Sensitivity of regularization model.

Figure 8: Sensitivity analysis of hyperparameters $\lambda$ and $\gamma$, as well as choice of estimators for subgroup density and local density on the synthetic dataset.

of covariate independence regularization too much, i.e. $\lambda > 0.5$, leads to a drop in performance. We hypothesize that this is due to the fact that the optimization focuses too much on minimizing covariate dependence, which can lead to very small subgroups with low exceptionality.

**Estimators.** We now compare the effect of using different estimators for the subgroup density $\hat{P}_{a,s}(Y)$ and the local conditional distribution $\hat{P}(Y \mid X = x, A = a)$. We compare the kernel density estimator (KDE) with a Gaussian Mixture Model (GMM), and a simple Gaussian. We display the results in Fig. 8c. In all settings, KDE slightly outperforms GMM, which in turn is better than a simple Gaussian. Especially in the demographic setting, using a Gaussian leads to a significant drop in performance. This is likely due to the fact that the simple Gaussian is not flexible enough to capture the true distribution of the target variable within the subgroup.

Regarding the local conditional distribution $\hat{P}(Y \mid X = x, A = a)$, we compare a Random Forest (RF) with a SVM and a neural network. We display the results in Fig. 8d. The results show now trend no clear winner across all settings. In the observational setting, NN performs best, while in the interventional and observational settings, RF has a slight edge. Ultimately, the choice of estimator for the local conditional distribution does not seem to have a significant impact on the performance of SUBCON in the synthetic experiments.

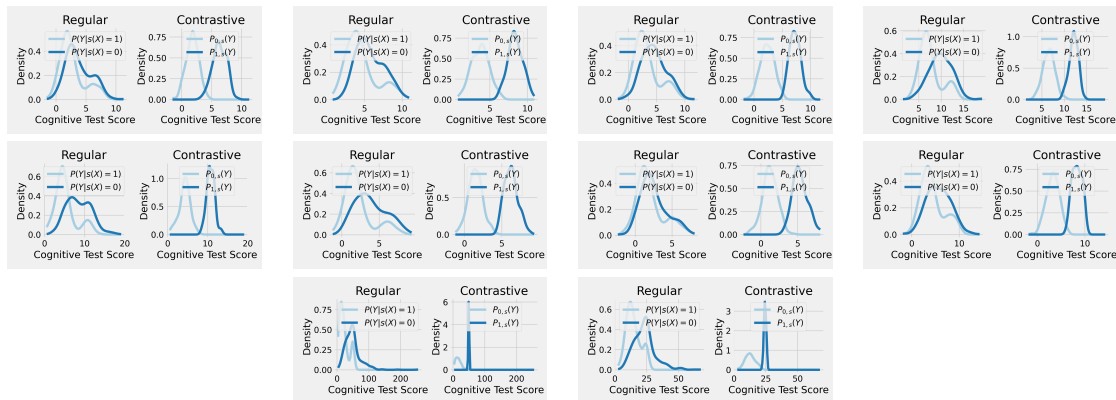

Figure 9: Subgroups discovered in the IHDP dataset by SUBCON.

# E    EXPERIMENTS: REAL WORLD

## E.1    IHDP

We further display the subgroup distributions discovered for each permutation of the IHDP dataset (Hill, 2011). We plot the subgroup distributions in Fig. 9. We list the rules for the discovered subgroups in Table 3.

| **Subgroup 1-5 Rules** | **Subgroup 6-10 Rules** |
|---|---|
| $-1.59 < X2\&X3 < 1.50\&X5 < 1.05\&X14 < 0.64\&X19 < 0.62$ | $-2.60 < X0\& -3.48 < X1\&X2 < 2.82\&X3 < 1.29\&X5 < 1.01\&X14 < 0.65\&X24 < 0.63$ |
| $X0 < 1.40\& -4.48 < X4\&X5 < 2.63\&X7 < 0.68\&X11 < 0.62\&X20 < 0.65$ | $X1 < 1.53\&X3 < 0.75\&X10 < 0.53$ |
| $X0 < 0.54\&X1 < 0.64\&X3 < 1.63\&X21 < 0.69$ | $X1 < 2.29\&X4 < 0.86\&X6 < 0.40\&X14 < 0.64\&X22 < 0.75\&X23 < 0.67$ |
| $X1 < 1.20\& -4.43 < X4\&X5 < 0.70\&X20 < 0.64\&X23 < 0.65$ | $0.24 < X17\&X19 < 0.85$ |
| $-0.21 < X3\& -1.78 < X5\&X6 < 0.40\&X13 < 1.47\&X19 < 0.58$ | $-1.24 < X2\&X3 < 1.42\&X4 < 0.66\&X11 < 0.54\&X18 < 0.66$ |

Table 3: Rules for each discovered subgroup in the IHDP dataset.

## E.2    COVID-19

We report the returned top subgroup from PYSUBGROUP, and two variants of SUBCON: SUBCON-max and SUBCON-min. SUBCON-max tries to maximize the distance $D$, whilst for SUBCON-min we set $D = -D_{JS}$, such that it minimizes the distance. In this section, we provide the top three subgroups found by SUBCON-max, SUBCON-min and PYSUBGROUP on the COVID-19 dataset. To obtain multiple subgroups using SUBCON, we iteratively run the optimization and remove those points, which have been identified as subgroup from the dataset after each run.

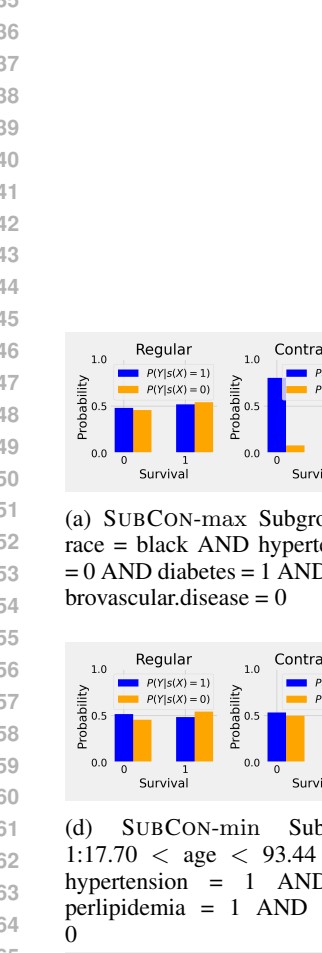

(a) SUBCON-max Subgroup 1: race = black AND hypertension = 0 AND diabetes = 1 AND cerebrovascular.disease = 0

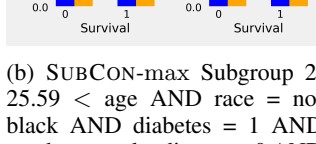

(b) SUBCON-max Subgroup 2: 25.59 < age AND race = not black AND diabetes = 1 AND cerebrovascular.disease = 0 AND hepatitis = 0 AND dementia = 0

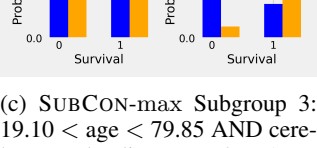

(c) SUBCON-max Subgroup 3: 19.10 < age < 79.85 AND cerebrovascular.disease = 0. AND chronic.kidney.disease = 0 AND copd = 1AND dementia = 0

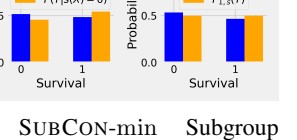
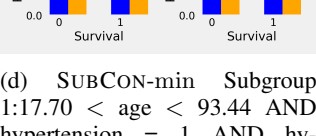

(d) SUBCON-min Subgroup 1:17.70 < age < 93.44 AND hypertension = 1 AND hyperlipidemia = 1 AND chf = 0

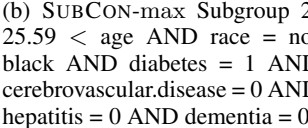
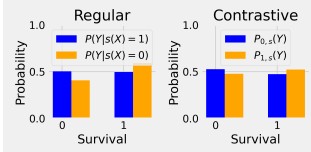

(e) SUBCON-min Subgroup 2: age < 71.00 AND hypertension = 1

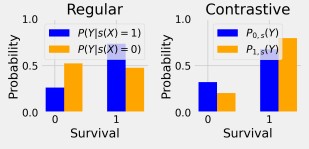

(f) SUBCON-min Subgroup 3: age < 76.40 AND hypertension = 1

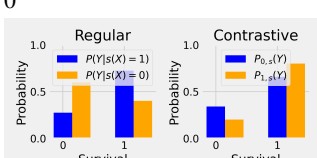
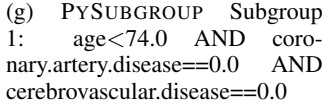

(g) PYSUBGROUP Subgroup 1: age<74.0 AND coronary.artery.disease==0.0 AND cerebrovascular.disease==0.0

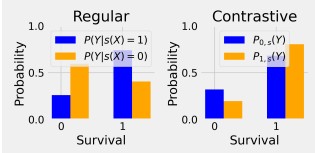

(h) PYSUBGROUP Subgroup 2: age<74.0 AND coronary.artery.disease==0.0 AND cerebrovascular.disease==0.0

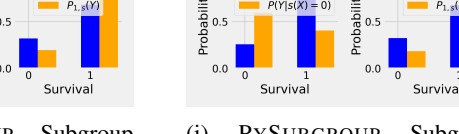

(i) PYSUBGROUP Subgroup 3: age<72.0 AND coronary.artery.disease==0.0 AND hepatitis==0.0

Figure 10: Top three subgroups discovered by SUBCON-max, SUBCON-min and PYSUBGROUP on the COVID19 ICU dataset Lambert et al. (2022).

### E.3 LIFE EXPECTANCY

We further present the results discovered on the WHO Life expectancy dataset (Ray, 2020). We differentiate between the developed nations ($A = 1$) and developing/least developed countries ($A = 0$).

SUBCON-max

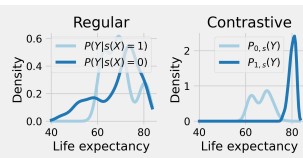 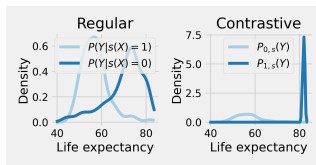 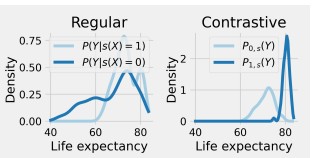

(a) SUBCON-max Subgroup 1: Under-five-deaths $<$ 112.36 AND Adult-mortality $<$ 254.76 AND Hepatitis-B $<$ 71.96

(b) SUBCON-max Subgroup 2: Under-five-deaths $<$ 172.39 AND Adult-mortality $<$ 493.28 AND Hepatitis-B $<$ 75.57 AND Measles $<$ 86.18

(c) SUBCON-max Subgroup 3: 2004.18 $<$ Year AND Infant-deaths $<$ 50.78 AND Adult-mortality $<$ 247.63 AND 32.50 $<$ Hepatitis-B AND Measles $<$ 86.46

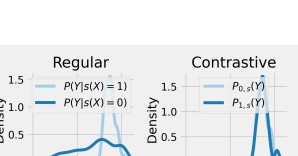 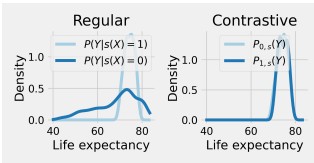 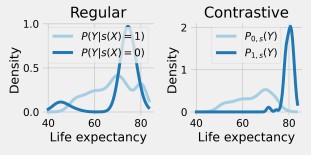

(d) SUBCON-min Subgroup 1: Adult-mortality $<$ 178.19 AND 32.09 $<$ Hepatitis-B AND 23.11 $<$ BMI AND 52.38 $<$ Diphtheria AND 4106.81 $<$ GDP-per-capita $<$ 92037.85 AND 1.06 $<$ Thinness-ten-nineteen-years $<$ 22.08 AND 2.16 $<$ Thinness-five-nine-years $<$ 22.49 AND 7.50 $<$ Schooling

(e) SUBCON-min Subgroup 2: 103.97 $<$ Adult-mortality $<$ 219.50 AND 9487.06 $<$ GDP-per-capita $<$ 24595.72 AND Thinness-ten-nineteen-years $<$ 20.01 AND Thinness-five-nine-years $<$ 22.55 AND 5.75 $<$ Schooling

(f) SUBCON-min Subgroup 3: Infant-deaths $<$ 83.39 AND Adult-mortality $<$ 436.42 AND Incidents-HIV $<$ 17.45

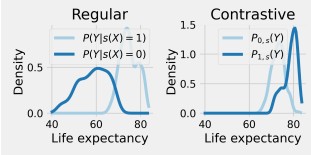 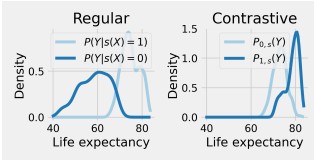 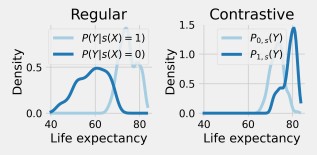

(g) PYSUBGROUP Subgroup 1: GDP-per-capita$>=$831.0 AND Under-five-deaths$<$41.0 AND Adult-mortality$<$247.14

(h) PYSUBGROUP Subgroup 2: Under-five-deaths$<$41.0 AND Adult-mortality$<$247.14

(i) PYSUBGROUP Subgroup 3: Under-five-deaths$<$41.0 AND Adult-mortality$<$247.14 AND Adult-mortality$<$429.31

Figure 11: Top three subgroups discovered by SUBCON-max, SUBCON-min and PYSUBGROUP on the WHO Life Expectancy Dataset.

