# OpenReview forum: "Contrastive Subgroups: Discovering Where Two Populations Differ, and Why"
_ICLR.cc/2026/Conference — Submitted to ICLR 2026_

### Official Review · Reviewer_ETqB · 2025-10-19

**Soundness:** 2
**Presentation:** 2
**Contribution:** 1
**Rating:** 2
**Confidence:** 4

**Summary:**

The authors propose a subgroup discovery method based on treatment effect estimation by employing the existing rule learner framework called SYFLOW.

For this goal, they have proposed formal notions called contrastive exceptionality and covariate dependence. However, these proposals are quite similar to the existing method for distributional treatment effect modifier discovery [Chikahara et al. UAI2022], which the authors do not even cite or mention.

**Strengths:**

- Problem setup is important: Many studies work on subgroup discovery based on treatment effect estimation, though the authors say “However, as of now, there does not exist a general definition or method to discover them in a principled way.” in Section 2.

- The authors discuss the relation between the proposed notions and some example toy causal graphs.

**Weaknesses:**

(A) No discussion on related work

The weakest point of this paper is the lack of related work discussion on subgroup discovery based on treatment effect estimation.
A prominent line of work is to discover **treatment effect modifiers** [1] from covariates, the features that explain why the treatment effects are different. The authors cite none of the related work on this topic [2-3] and just say that “However, as of now, there does not exist a general definition or method to discover them in a principled way.” in Section 2.

A serious issue is that the first contribution of this paper “(formalize the notion of contrastive subgroups)” is **NOT novel**, as **the proposed notion of contrastive exceptionality seems very close** (and the underlying idea is almost identical) *to the existing feature importance measure** based on conditional potential outcome distributions [3].

Due to the lack of related work discussion, I could not find the novelty, significance, and originality of this paper, thus I cannot recommend the acceptance to such a top conference as ICLR.

> [1] Rothman, Kenneth J., Sander Greenland, and Timothy L. Lash. Modern epidemiology. Vol. 3. Philadelphia: Wolters Kluwer Health/Lippincott Williams & Wilkins, 2008.

> [2] Zhao, Qingyuan, Dylan S. Small, and Ashkan Ertefaie. "Selective inference for effect modification via the lasso." Journal of the Royal Statistical Society Series B: Statistical Methodology 84.2 (2022): 382-413.

> [3] Yoichi Chikahara, Makoto Yamada, Hisashi Kashima. Feature Selection for Discovering Distributional Treatment Effect Modifiers. UAI, 2022.


(B) Weakness of rule learning approach

Another weakness of this work is the lack of evaluating statistical significance of detected subgroups. Unlike the statistical-testing-based approaches [2-3], the authors learn interpretable, tree-based models. Although this approach is popular in existing work (e.g., [4]; though the authors do not mention or cite again), such approach cannot evaluate the statistical significance of the inferred results.

Hence, the learned subgroups are easily changeable due to the noise in empirical distribution, and the robustness is doubtful. Additional experiments for confirming the noise robustness will be necessary to claim the technical soundness.


> [4] Falco J. Bargagli-Stoffi, Riccardo Cadei, Kwonsang Lee, Francesca Dominici. Causal Rule Ensemble: Interpretable Discovery and Inference of Heterogeneous Treatment Effects. arXiv


(C) Inappropriate citations or related work reference on fairness

The description on algorithmic fairness (based on causality) seems inappropriate.

- The proposal of [Dwork et al., 2012] is **NOT** demographic parity, but individual fairness definition. These notions are totally different. The citation is completely inappropriate.

> These methods provide notions to quantify disparity between two populations, but they leave open where that disparity arises and why their distributions diverge.

- The task of elucidating why the disparity arises is often called “discrimination discovery”, and there are several methods for this task. A pioneer work would be [5-6], which are **NOT** cited again in this paper.

> [5] Lu Zhang, Yongkai Wu, and Xintao Wu. Situation Testing-Based Discrimination Discovery: A Causal Inference Approach. IJCAI, 2016.

> [6] Zhang, Lu, Yongkai Wu, and Xintao Wu. "Causal modeling-based discrimination discovery and removal: Criteria, bounds, and algorithms." IEEE Transactions on Knowledge and Data Engineering 31.11 (2018): 2035-2050.

**Questions:**

N/A

---

> ### Author Response · Authors · 2025-11-20
>
> We thank the reviewer for highlighting the work on treatment effect modifiers. These methods (e.g., Zhao et al., Chikahara et al.) identify subsets of **features** that explain treatment-effect heterogeneity, e.g. “Diabetes affects CATE with p value $0.0075$”. In contrast, the focus of our work is to discover subgroups of individuals with an exceptional disparity, i.e. subsets of **samples** defined through rules such as “large disparity in mortality for patients with Diabetes = 1 & Hypertension = 0 ”.
>
> While both approaches quantify divergence by contrasting the distribution of both populations, Chikahara et al. *measure* variability when conditioning on a *single* covariate as per $$Var_{x \sim X_m}[D(P(Y\mid X_m=x, A=0), P(Y\mid X_m=x,A=1)]\;,$$ whereas we *learn* a subgroup membership function by optimizing divergence through $$\arg \max_s D(P_{0,s}(Y),P_{1,s}(Y))\;.$$ Thus, the two approaches address related, complementary problems. The presence of substantial treatment-effect variability for a single covariate can motivate the search for more specific, multi-condition subgroups exhibiting pronounced disparity, which is exactly the objective we formalize and address.
>
> - **Discrimination discovery**: The works of Zhang et al. address an important, but complementary task. The 2016 paper detects **individual-level** discrimination by comparing a decision, e.g. loan approval, for an individual to matched counterparts with similar characteristics measured by a pairwise distance function.  They can thus state for every individual whether there is discrimination, but *unlike explicit subgroup discovery*, they do not describe these characteristics in a human-interpretable way. Zhang et al. (2019) focus on *removing discrimination* but again do *not return subgroup descriptions*. We are happy to clarify this distinction in the fairness discussion and include these citations, as they make clear that SUBCON is complementary: it identifies where disparities manifest, while discrimination discovery methods investigate if they occur.
>
> - **Statistical parity** We kindly but strongly disagree with the critique that Dwork et al. is inappropriate; Definition 3.1 defines statistical parity on the distribution level. Nevertheless, we are happy to exchange it for another reference that specifically defines and discusses statistical parity.
>
> - **Robustness and statistical significance**: While our method is not testing-based and hence does not provide formal statistical significance guarantees, we conducted an additional robustness experiment by progressively adding up to 95 uninformative noise features to the dataset. The results, reported in the table below, show that SUBCON maintains stable performance (F1 ranging from 0.82 to 0.87) and only moderate increases in computation time. This indicates that the discovered subgroups are resilient to noise in the empirical distribution and not overly sensitive to feature perturbation. We also note that, if required for whichever reason, statistical significance can always be determined using a holdout set or through permutation testing (e.g. see Duijvestijn & Knobbe, 2011).
>
>    | #Noise features | 5 | 20 | 45 |70 | 95|
>    |-------|------|------|-------|-------|-------|
>    | **$F_1$** | $0.87$ | $0.82$ | $0.85$ | $0.87$ | $0.82$ |
>    | **Runtime (s)** | $21$ | $31$ | $55$ |$57$ |$62$ |
>
> - **Tree-based CATE estimation**: Finally, regarding rule- or tree-based CATE estimation approaches such as the Causal Rule Ensemble, we already mention and compare against tree/forest-based CATE models, which can yield subgroups but are primarily designed to produce accurate individual-level treatment effect estimates. These models optimize predictive accuracy rather than contrastive disparity and do not incorporate our notion of contrastive exceptionality or covariate independence. SUBCON instead directly seeks subgroups with maximal contrast between populations, providing interpretable descriptions specifically for disparity discovery rather than general CATE estimation.
>
> The three domains raised by the reviewer, treatment effect modifier analysis, discrimination discovery, and tree-based CATE estimation, each highlight the relevance of identifying regions of disparity. However, none of these approaches explicitly addresses the task of discovering these contrastive subgroups. This underscores both the utility of our contribution and its distinction from existing work. We again thank the reviewer for their careful evaluation and will incorporate the suggestions to further strengthen the contextual framing of our contribution for the reader.
>
> Duivesteijn, Wouter, and Arno Knobbe. Exploiting false discoveries--statistical validation of patterns and quality measures in subgroup discovery. IEEE 11th International Conference on Data Mining (2011).

---

> > ### Comment · Reviewer_ETqB · 2025-11-26
> >
> > I thank the authors for their detailed response. However, I feel that this paper requires a major revision to be accepted to such a top conference as ICLR. Therefore, I will keep my rating of "2: rejection".
> >
> > The reason why I pointed out the absence of related work discussion is that we cannot evaluate the significance and novelty of this work. For this reason, I have cited numerous related work references to encourage the authors to elaborate on the significance. Unfortunately, however, they simply describe some superficial (and seemingly minor) differences between their work and the existing work (I understand most of them because it is I who identified these references) and fail to convince me of the significance of their work.
> >
> > In the authors' future revision, I hope they will seriously consider my comments to elaborate on the significance of this work. For instance, this includes the illustration of some real-world scenarios where only their proposed approach can be applied, but none of the existing frameworks (e.g., treatment effect modifier detection or rule-based models) can address. Such elaboration will require a major revision of e.g., Introduction, and I cannot imagine the revised one from the current version.
> >
> > > We kindly but strongly disagree with the critique that Dwork et al. is inappropriate; Definition 3.1 defines statistical parity on the distribution level. Nevertheless, we are happy to exchange it for another reference that specifically defines and discusses statistical parity.
> >
> > To my understanding, Definition 3.1 in Dwork et al. is introduced to identify the aspects in which statistical parity is inappropriate, before describing their main proposal, namely, the criterion of individual fairness. This is why I say that the authors' citation is completely inappropriate.
> >
> > If I were the co-authors, I would recommend the pioneering work that aims to train a predictive model that achieves statistical parity.  For instance,
> >
> > > Kamishima, T., Akaho, S., Asoh, H., & Sakuma, J. Fairness-aware classifier with prejudice remover regularizer. In Joint European conference on machine learning and knowledge discovery in databases, 2012.
> >
> > is a widely known reference in the community of algorithmic fairness; other survey papers on the algorithmic fairness, including the introduction of statistical fairness formulation, could be OK. Since I am **NOT** the co-authors of this work, I hope that the authors seriously seek their appropriate references by themselves, before asking their reviewer.

---

### Official Review · Reviewer_b8UN · 2025-10-28

**Soundness:** 3
**Presentation:** 3
**Contribution:** 3
**Rating:** 4
**Confidence:** 3

**Summary:**

This paper proposes SUBCON, a differentiable framework for contrastive subgroup discovery, a novel task that aims to identify subsets where the target distribution differs significantly between groups (e.g. treatment v.s. control).  The method is a combination of exceptionality, generality, and covariate independence. To make these discovered subgroups actionable, the authors provide conditions under which the discovered subgroups allow to make causal inferences. Experiments on simulated, IHDP, and COVID-19 datasets show that SUBCON reliably recovers meaningful contrastive subgroups, outperforming both traditional subgroup discovery and causal-effect baselines.

**Strengths:**

1. Novelty: The paper introduces a new task, contrastive subgroup discovery, which is a largely unexplored research direction. Beyond defining this novel problem, it also proposes a concrete differentiable solution. The contribution is conceptually original, and the novelty is well established.
2. Significance: The proposed task has potential implications for multiple downstream domains, such as fairness analysis. From my perspective, the formulation could be useful not only for real-world data mining applications but also for theoretical studies on causal and subgroup structures.
3. Clarity: The paper is generally well written and easy to follow. Technical details are presented clearly and self-contained, and the figures (e.g. Figure 1 and 2) are intuitive and informative.

**Weaknesses:**

1. The following statement could be improved:
“By ensuring that the features do not locally influence the target variable, we can be more confident that the observed differences are not due to a shift in features caused by $A$, but instead directly driven by $A$ itself.”
Minimizing the so-called “covariate dependence” seems to block both pathways $A$→$X$→$Y$ and $A$←$X$→$Y$, so it may not be rigorous to claim “features caused by $A$”.
2. I would be interested in seeing more discussion or experimental illustrations of potential downstream applications, such as fairness analysis, to demonstrate the broader utility of SUBCON.
3. While the proposed idea of contrastive subgroup discovery is interesting and well-motivated, much of the technical implementation (e.g. differentiable rule learner and optimization design) builds directly on prior work (e.g. Xu et al., SyFlow). Consequently, the main novelty lies in the problem framing rather than algorithmic innovation, and the paper explicitly focuses on the former.

**Questions:**

1. Is it possible (though not strictly necessary) to provide some theoretical guarantees that the proposed learning procedure can indeed recover the (ground-truth) subgroup indicator? For example, a learning-theoretic bound or convergence analysis.
2. As I understand, the same hyperparameter settings were used for both real-world and simulated datasets. Are these hyper-parameter selection rules generally suitable across different settings, or mainly for the small-graph cases in your simulations? It would be interesting to see experiments on larger and more complex datasets.
3. I wonder how the performance of SUBCON scales with the dimensionality of $X$ and the number of subgroups involved. Have you evaluated the method’s robustness when varying the feature dimensionality? For instance, higher-dimensional $X$ may correspond to more complex subgroup structures and potentially harder optimization.
4. For the real-world datasets, could you provide more scientific justification or evidence for the meaningfulness of the discovered subgroups? In other words, do the identified subgroups correspond to interpretable or actionable patterns in the underlying domain?

The paper introduces a novel and meaningful problem setting with an interesting formulation. While the current version still lacks clarity and sufficient empirical validation, I believe the work has value for the ICLR community. I lean towards a weak reject at this stage, but I acknowledge the potential impact of this work and remain open to increasing my score based on the authors’ rebuttal.

---

> ### Author Response · Authors · 2025-11-20
>
> Dear Reviewer,
> Thank you for your constructive comments. We would like to address the perceived weaknesses first.
> - **Clarification of Covariate Independence**: Thank you for raising this point. The intention behind covariate independence is to block paths from $X \to Y$ to remove the covariates effect on the target variable, regardless of whether $X$ is a cause or an effect of $A$. This is especially relevant in chains such as $A \to X \to Y$, where otherwise we may falsely attribute effects to $A$ that are realized through proxy by $X$. Minimizing $\mathcal{C}(s)$ mitigates this issue. In the alternate case, $A \leftarrow X \rightarrow Y$, by construction there are no features $X$ caused by $A$ that influence $Y$, so that no matter the causal structure with $\mathcal{C}(s)=0$, we can be sure that any disparity does not come from an intermediate $X$. We will modify the manuscript accordingly to more clearly reflect this reasoning.
> - **Downstream Applications**: Contrastive subgroup discovery can offer highly valuable insights across various domains. For instance, we are currently exploring the use of SUBCON in gender-specific medicine to identify demographic subgroups in which drug efficacy differs substantially. In this context, the discovered subgroups serve as hypothesis-generating insights to launch further studies to validate the findings and investigate the underlying biological mechanisms. As you may appreciate, conducting and evaluating full application-focused studies requires a dedicated research effort beyond the scope of this work. Nevertheless, in line with your suggestion, we will include a discussion section in the revised manuscript outlining practical use cases to guide potential users.
> - **Prior Work**: We agree that our primary contribution lies in formulating the novel task of contrastive subgroup discovery and analyzing its causal implications. However, we would like to emphasize that addressing this task required several non-trivial technical developments beyond reusing existing methodology. While our approach adopts the differentiable rule-learning backbone from Xu et al., we introduce a novel covariate dependence regularization procedure and a different methodology for computing contrastive instead of conventional exceptionality.

---

> > ### Author Response · Authors · 2025-11-20
> >
> > Regarding your questions:
> > - **Guarantees**: The setting, where a ground-truth subgroup indicator is present, represents idealized conditions that are best suited for evaluating subgroup discovery performance. In contrast, real-world data typically contains multiple subgroups exhibiting varying levels of disparity rather than a single ground truth. Under such conditions, the task focuses on identifying a *globally optimal* subgroup. In the general case, however, classic subgroup discovery is known to be NP-hard [1]. That said, branch-and-bound approaches can successfully find optimal solutions for smaller-scale problems [2]. We therefore believe that analogous optimistic estimators and algorithms could be developed for contrastive subgroup discovery, particularly for certain distance measures such as mean-based distribution distances.
> > - **Hyperparameters**: All results reported in the paper, both for synthetic and real-world datasets, were obtained with the same set of hyperparameters. This configuration yielded satisfactory performance across all scenarios evaluated. As part of this rebuttal, we additionally analyze the behavior of SUBCON under higher-dimensional settings, as detailed below.
> > - **Dimensionality**: In terms of runtime, the differentiable component SUBCON scales linearly with both the number of features and the number of samples. To test the robustness of our method under increasing feature dimensionality, we consider two separate settings: adding uninformative, noise features, and adding informative, causal features. We report the respective results in the two tables below. SUBCON remains largely robust to the inclusion of uninformative features. However, performance deteriorates with increasing numbers of informative features, with a marked decline once the number exceeds 20 features that are direct causes of $Y$ and also influence $A$.
> >
> >    | #Noise features      | 5 | 20 | 45  |70 | 95|
> >    |-------|------|------|-------|-------|-------|
> >    | **$F_1$**     |    $0.87$  |   $0.82$   |   $0.85$    | $0.87$ | $0.82$ |
> >    | **Runtime (s)** | $21$     |  $31$    |   $45$    |$57$ |$62$ |
> >
> >    | #Informative features      | 5 | 10 | 20   | 50|
> >    |-------|------|------|-------|-------|
> >    | **$F_1$**     |   $0.87$   | $0.86$     |  $0.50$     |  $0.19$ |
> >    | **Runtime (s)** |    $15$  | $18$     | $25$      | $36$ |
> > - **Meaningfulness**: To address your question regarding the scientific justification and interpretability of the discovered subgroups in real-world data, we focus on the COVID-19 mortality subgroup presented in Section 7.2. There, SUBCON identifies a subgroup in which the mortality rate of men is substantially higher, characterized by
> >  “Race=black & Diabetes=1 & Cerebrovascular Disease = 0 & Hypertension = 0”. To start, we analyze the results of an Italian study with a specific focus on gender differences in COVID-19 [1]. The study conducts a Chi-squared test between each comorbidity and mortality and reports the following $p$-values for women and men:
> > | | Male | Female |
> > |-------|-------|-------|
> > |Diabetes |$0.004$ | $0.835$|
> > |Cerebrovascular Disease |$<0.001$ |$0.017$ |
> >
> >    This analysis reveals a significant association between diabetes and mortality in men, but not in women, whereas cerebrovascular disease (CVD) appears as a risk factor for both. In light of this, our subgroup can be interpreted as follows: including Diabetes likely drives the subgroup’s contrastive exceptionality for males, while excluding CVD helps remove its influence via covariate dependence regularization.
> >
> >    Regarding ethnicity, an American study reports a substantial gap in age-adjusted mortality between Black males (higher) and Black females [4]. Our findings align with this pattern, although it remains an open question which specific interactions between comorbidities, sex, and ethnicity drive the observed disparities.
> >
> >    Overall, these connections to existing clinical evidence suggest that the discovered subgroup corresponds to an interpretable and potentially actionable pattern in the medical domain. We view contrastive subgroup discovery primarily as a hypothesis-generation tool that highlights such disparity patterns, which may be difficult to uncover through manual expert analysis alone, and thus can inform targeted follow-up studies.

---

> > > ### Author Response · Authors · 2025-11-20
> > >
> > > [1] Boley, Mario, and Henrik Grosskreutz. Non-redundant subgroup discovery using a closure system. In Joint European Conference on Machine Learning and Knowledge Discovery in Databases (2009).
> > >
> > > [2] Grosskreutz, Henrik, Stefan Rüping, and Stefan Wrobel. Tight optimistic estimates for fast subgroup discovery. Joint European conference on machine learning and knowledge discovery in databases (2008).
> > >
> > > [3] Agodi, Antonella, et al. Gender differences in comorbidities of patients with COVID-19: An Italian local register-based analysis. Heliyon 9.7 (2023).
> > >
> > > [4] Rushovich, Tamara, et al. Sex disparities in COVID-19 mortality vary across US racial groups. Journal of General Internal Medicine 36.6 (2021).

---

### Official Review · Reviewer_1G57 · 2025-11-01

**Soundness:** 3
**Presentation:** 3
**Contribution:** 3
**Rating:** 6
**Confidence:** 3

**Summary:**

This paper introduces contrastive subgroups—subsets of data where two populations with similar characteristics show significantly different outcomes. It formalizes this problem, provides causal conditions for valid interpretation, and proposes SUBCON, a gradient-based method to discover such subgroups. Experiments show SUBCON can reveal meaningful treatment heterogeneity and demographic disparities that standard subgroup discovery or causal models miss.

**Strengths:**

This paper’s main strength is that it identifies and formalizes an important but previously under-addressed problem: finding subgroups where two populations differ locally, rather than only measuring global disparities or blindly searching for exceptional patterns. The formulation is elegant and bridges subgroup discovery with causal inference in a principled way, making the results not just descriptive but potentially actionable.

* Methodologically, the paper offers a well-constructed objective that balances statistical exceptionality, support, and confounding control, backed by clear causal interpretations. The SUBCON algorithm is thoughtfully engineered, combining differentiable rule learning with divergence-based objectives and regularization for causal validity.

* The experimental evaluation is comprehensive, spanning controlled synthetic settings with known ground truth as well as semi-synthetic and real-world datasets. The results compellingly show that existing subgroup discovery and causal methods each miss key patterns that the proposed method identifies. The examples, especially in fairness and clinical-style settings, emphasize the practical relevance and interpretability of the discovered subgroups.

* The paper is clearly written and motivates the problem well, using intuitive examples to illustrate why contrastive subgroup discovery matters. Overall, this is a strong and timely contribution that introduces a meaningful new problem and provides an effective and interpretable solution.

**Weaknesses:**

While the paper makes a strong contribution, there are a few limitations. The method relies on accurate density and conditional distribution estimation, which may be challenging in higher-dimensional or noisy settings. The pipeline also has several components (e.g., temperature annealing, re-estimating densities), making it more complex to implement and tune compared to tree-based causal approaches. Finally, the method currently focuses on discovering one subgroup at a time, and extending it to systematically identify multiple subgroups would further increase its practical utility.

**Questions:**

1. How sensitive is SUBCON to inaccuracies in density and conditional distribution estimation, particularly in higher-dimensional settings or smaller datasets?

2. Could the authors provide more guidance or ablation results on tuning the key hyperparameters (e.g., temperature schedule, λ, γ)? Are there practical heuristics to simplify training?

3. Do the authors envision a principled extension for discovering multiple contrastive subgroups simultaneously, rather than sequentially masking previously found groups?

4. What is the computational scalability of SUBCON for large-scale tabular data (e.g., 100k+ samples or dozens of features)? Have the authors benchmarked runtime or memory overhead?

---

> ### Author Response · Authors · 2025-11-20
>
> Dear Reviewer,
> Thank you for your constructive and valuable feedback. We address your main points in detail below.
> - **Effect of Estimators**: In Appendix D1, we present a set of experiments that investigate the impact of selecting different estimators. Regarding the density estimators, our findings indicate that on the synthetic datasets, KDE and GMM-based estimators perform equally well, whereas a simple Gaussian estimator leads to reduced accuracy in recovering the ground-truth subgroup. We also vary the kind of model used to model the conditional density/mean, where we again find no conclusive advantage between using an SVM, random forest, or MLP. Across these variations, SUBCON’s performance remains stable in all three causal settings, suggesting that the method does not hinge on a specific choice of estimator. We will add clear pointers to these results in the main experimental section.
> - **Hyperparameter Tuning**: Similarly, we also perform a systematic sensitivity analysis of the hyperparameters γ and λ. This analysis, included in Appendix D.1, shows that the method is stable across a range of $\gamma \in [0.1, 0.3]$ and across a broad region of λ up to $0.5$. Regarding the temperature, we provide a description of our scheduler, including parameters in Appendix B.3. Our findings are that the linear schedule works well in all tested cases and need not be modified.
> - **Multiple Subgroups**: Finally, you mention the limitation that the method discovers one subgroup at a time. We agree that this is a natural direction for extension of the framework.  In this work, however, our focus is on establishing the field of contrastive subgroup discovery, both theoretically and through a practical methodology.
> To generalize towards sets of contrastive subgroups, we think that a global approach based on a mixture of subpopulations offers the most promising and principled path forward. This, however, introduces several theoretical and practical challenges, which we look forward to addressing in future work.
> - **Scalability**: SUBCON relies on continuous optimization and introduces an overhead that scales linearly with both the number of features and the number of samples. Our density estimator for continuous data (Gaussian KDE) has a computational complexity of $O(n^2)$. In Appendix C2, we assess runtime performance by scaling the number of features $d$ in synthetic data up to 100.  In this setting, SUBCON incurs a small overhead compared to SyFlow due to additional regularization and the need to estimate densities for both populations, with a single training run taking approximately 20 seconds on a laptop for 100 variables.
>
>    We now extend this analysis in an experiment with up to 10,000 samples with 10 fixed variables, reporting the runtime and $F_1$ scores. Increasing the number of samples improves the $F_1$​ score and primarily impacts runtime via the density estimation step, reaching 6 minutes per subgroup for 10,000 samples. Performance could potentially be improved through subsampling strategies or more scalable density estimators.
>    |       | 1,000 | 5,000 | 10,000 |
>    |-------|------|------|-------|
>    | **$F_1$**     | $0.84$     | $0.86$     |   $0.92$    |
>    | **Runtime (s)** |  $10$    |  $60$    |   $330$    |

---

### Official Review · Reviewer_SWta · 2025-11-03

**Soundness:** 3
**Presentation:** 4
**Contribution:** 4
**Rating:** 4
**Confidence:** 3

**Summary:**

This paper presents an approach to finding "contrastive subgroups" - areas of the input space where two populations differ in outcomes according to a label Y. They present the criteria that these subgroups must hold, and connect some of these criteria to causal interpretations. Empirically, they demonstrate that this method is able to find contrastive subgroups in synthetic and semi-synthetic data and can also be used in some cases for causal effect estimation.

**Strengths:**

- very interesting framing and question that hasn't seen that much work
- connections to causality are nice and useful
- experiments are convincing to me that the method is more successful at finding contrastive subgroups than existing alternatives

**Weaknesses:**

- I think the 3 criteria may miss a specific corner case: consider a subgroup S that consists of 2 disjoint subsets S1 and S0. S1 contains only A=1 and S2 contains only A=0. Then, generality is non-zero, and we can have both covariance independence and exceptionality by assigning Y=1 on S2 and Y=0 on S0 uniformly. This seems as though it does not satisfy the notion of contrastive subgroups that you want since there is 0 overlap between the two groups. In particular, this may be an issue for the approach as S is implemented in a more flexible way than the highly-parameterized approach used here. I think the contribution here is still useful but it does knock the utility of the theoretical contributions down for me - unless I'm missing something, in which case I would love to be corrected
- additionally, this specific corner case may cause some of the causal reasoning, such as in Prop 1, to fail, since standard positivity notions are not satisfied
- the generality metric is odd to me - it's not clear that once we get sufficiently far from 0, we necessarily want to bias towards larger subgroups
- also on generality, in Eq 6 I don't see why it makes sense to multiply generality by exceptionality - more intuition here would be helpful! At the moment it seems arbitrary
- Eq 16: a little unclear, is this supposed to  be JS divergence? what is Q(Y)?
- missing some clarity on the PEHE experiments - they're interesting but I don't totally understand how the subgroup maps onto a causal effect estimate: is there some assumption made about what subgroup you're going to find? I understand the basic causal connections mentioned previously, but I'd imagine you could get very different causal effects by finding different subgroups

**Questions:**

- would like to see the corner case from the Weaknesses section addressed, or tell me why I'm wrong about it
- two questions on generality: seems odd to maximize (rather than threshold above some number), and seems odd to multiply it with exceptionality as in Eq 6
- more clarity on PEHE experiments

Happy to increase my score if these two things are addressed.

---

> ### Author Response · Authors · 2025-11-20
>
> Dear Reviewer,
>
> Thank you for your constructive comments. Below we address your main points.
> - **Corner case**: The scenario you lay out does indeed *satisfy* our defined criteria of exceptionality and generality without any measured covariate interference. However, it *violates* the positivity condition $0<P(A \mid X=x)<1$ that is required for $X$ to serve as a valid backdoor adjustment variable according to Eq. (7).  Hence, positivity of the treatment/group assignment is implicitly required in Proposition 1, by which the given scenario is in fact handled correctly. We completely agree it is a good idea to explicitly state this condition, and will do so in the updated manuscript.
>
>
>    In practice, positivity violations can pose challenges, particularly in smaller high-dimensional datasets, where most individuals are unique (i.e., each $x$ occurs mostly once with $a=0$ or $a=1$ but not multiple times with different $a$). Our subgroup-based approach mitigates the challenge in conditioning on individual $x$ by estimating group-level distributions over rule-defined subsets of the data. Nevertheless, to avoid subregions of the covariate space where only one treatment group is supported, a preprocessing step (potentially involving regular subgroup discovery) would be required to guarantee positivity. As designing such a procedure is nontrivial and beyond the scope of this work, we look forward to addressing it in future research.
> - **Generality**: Our use of an exceptionality term weighted by size follows the standard pattern in subgroup discovery, where “support times exceptionality” type measures are common (see for example Herrera et al., 2011 [1]). Maximizing such an objective intentionally biases toward larger subgroups, but helps avoid the degenerate extreme where, in the absence of a threshold, a single highly divergent point becomes ‘optimal’. On the other hand, using a threshold induces a bias towards smaller subgroups up to the threshold. For example, when optimizing the mean difference, dropping points that are closer to the mean is always preferable and thus steers optimization towards smaller subgroups. Which bias is preferred in practice thus depends on the user. Within our differentiable framework, using either scheme requires only minimal modification to the loss function.
> - **Eq. 16**: For the exceptionality, we indeed use the Jensen-Shannon divergence, comparing the two outcome distributions inside the subgroup against their mixture distribution. In the notation of the paper, the mixture is denoted $M$, not $Q$, and is defined just above. We will correct this typographical error in the revised version.
> - **CATE Experiment**: We use SUBCON to discover a subgroup in the IHDP dataset, distinguishing between the treated and untreated, i.e. A in this case is the treatment indicator T. We estimate the treatment effect within the subgroup as $E[Y|T=1,s(X)=1]-E[Y|T=0,s(X)=1]$ and outside it as $E[Y|T=1,s(X)=0]-E[Y|T=0,s(X)=0]$, in analogy to causal trees which do this for every leaf. Across the ten IHDP replications, we find that SUBCON consistently identifies a subgroup for which the in-subgroup estimator achieves improved PEHE compared to other explainable baselines such as causal trees and causal forests. We will clarify this construction and the interpretation of the PEHE numbers in the revised text.
>
> [1] Chapter 3 of Herrera, F., Carmona, C. J., González, P., & Del Jesus, M. J. (2011). An overview on subgroup discovery: foundations and applications. Knowledge and information systems, 29(3), 495-525.

---

### Meta-Review · Area_Chair_3FYw · 2026-01-08

**Summary:**

My recommended decision is based primarily on concerns about positioning/related work and novelty that remained outstanding after rebuttal. However, I do recognize the generally positive assessments of the paper’s framing and problem setup from multiple reviewers.

Reviewer SWta found the paper interesting and appreciated the framing of contrastive subgroups, indicating potential willingness to increase their score, but raised presentation/clarity questions (e.g., interpretation of the PEHE experiments and discussion of corner cases). In my reading, the rebuttal addresses many of these points, though the discussion did not give a clear roadmap on how that these clarifications were incorporated into a revised manuscript.

Reviewer 1G57 was similarly positive and leaned toward acceptance. Their concerns were comparatively minor and appear largely addressed in the rebuttal (mostly by pointing to the existing appendix).

Reviewer b8UN noted weaknesses but also indicated willingness to increase their score with clarifications (though remained silent post-rebuttal). The authors addressed several issues (e.g., precision about statements and hyperparameter settings) largely by again pointing to relevant appendices.

The most contentious review was ETqB, who maintained a reject score after rebuttal, emphasizing insufficient novelty relative to prior work and missing/incorrect related-work framing. I agree in particular with ETqB’s point that citing "Fairness Through Awareness" as “introducing statistical parity” is inaccurate in emphasis: the paper discusses statistical parity while in route to show its limitations and motivate an alternative notion (“fairness through awareness”). Statistical parity has appeared in prior work.

More broadly, I believe the manuscript misses a substantial slice of the multi-group fairness literature on systematically identifying subpopulations where a model behaves poorly (e.g., multiaccuracy and multicalibration), including work by Kim et al. and Hébert-Johnson et al. (see "Calibration for the (Computationally-Identifiable) Masses" and "Multiaccuracy: Black-Box Post-Processing for Fairness in Classification"). While the goals are not identical (discovering where two populations differ vs. auditing/repairing a predictor across subgroups), the “contrastive subgroup” notion is close enough that clearer positioning is necessary. More concerningly, this indicates a deeper disengagement with the more recent fairness literature, echoing points raised by ETqB.

Overall, the results are interesting, but the presentation and related-work positioning issues, especially relative to multi-group fairness and related literature pointed in the reviews, make this better suited for revision and resubmission after an additional round of polishing and repositioning. I believe this will make an excellent paper after an additional round of review.

**Reviewer Concerns:**

### Concerns largely addressed in rebuttal (or appear addressed):

* SWTta: Many of the clarity questions (including around PEHE interpretation and corner-case discussion) appear to be addressed in the rebuttal, though it is not clear how these changes will be reflected in an updated paper draft.

* 1G57: Minor concerns; the rebuttal/appendix pointers seem to address them.

* b8UN: Several concrete issues (precision of statements, which hyperparameters were used, where to find specific details) were addressed, often by pointing to appendix sections the reviewer may have missed.

### Concerns still outstanding (in my view):

* ETqB: The reviewer remained unconvinced on novelty/overlap with prior work and on breadth/accuracy of related work. I agree that the paper needs a clearer, more accurate positioning, especially around fairness-related citations and adjacent literature.

**Reviewer Scores:**

* SWta: Likely a small increase if they had participated fully in discussion, given their stated openness and that many clarity points appear addressed.

* 1G57: Likely unchanged (already positive), or at most a small increase, since remaining issues were minor.

* b8UN: Unclear, but plausibly a small increase given the rebuttal’s appendix pointers and clarifications. However, their concern that much of the implementation builds on prior work suggests any increase may be limited due to novelty concerns.

 * ETqB: Unchanged (reject). The reviewer explicitly maintained rejection post-rebuttal, and I agree the related-work/positioning issues they highlighted warrant another revision cycle.

---

### Decision · Program_Chairs · 2026-01-26

Reject